# Can Dietary Supplements Support Muscle Function and Physical Activity? A Narrative Review

**DOI:** 10.3390/nu17213495

**Published:** 2025-11-06

**Authors:** Louise Brough, Gail Rees, Lylah Drummond-Clarke, Jennifer E. McCallum, Elisabeth Taylor, Oleksii Kozhevnikov, Steven Walker

**Affiliations:** 1School of Food Technology and Natural Sciences, Massey University, Palmerston North 4442, New Zealand; 2School of Biomedical Sciences, University of Plymouth, Drake Circus, Plymouth PL4 8AA, UK; 3School of Life Sciences, University of Sussex, Brighton BN1 9RH, UK; lylahdc@stgmed.com; 4St Gilesmedical London & Berlin, Suite 616, The Shepherds Building, Charecroft Way, West Kensington, London W14 0EE, UK; jennifer.mccallum@stgmed.com (J.E.M.); e.p.taylor@dundee.ac.uk (E.T.); steven.walker@ucl.ac.uk (S.W.); 5School of Life Sciences, Carnelley Building, University of Dundee, Dundee DD1 4HN, UK; 6MedibiotiX GmbH, Dr.-Reckeweg-Str. 2-4, 76532 Baden-Baden, Germany; oleksii.kozhevnikov@medibiotix.net; 7Medical School, University College London, 74 Huntley St, London WC1E 6DE, UK

**Keywords:** nutrition, dietary supplementation, muscle function, muscle performance, muscle mass, sarcopenia, inflammation, microbiome, review

## Abstract

Dietary supplementation is commonly used by athletes to gain muscle mass, enhance performance, and improve recovery. Most adults engage in insufficient physical activity. Yet healthy muscles are also critical for activities of daily living (ADLs), maintaining a good quality of life and positive ageing. There is growing interest in whether dietary supplementation is of value, particularly among subgroups such as the occasionally active, the ill and elderly, and peri- and menopausal women. By focusing on function, performance, mass and strength, ADLs, exercise-induced muscle damage and delayed onset muscle soreness, this review sought to examine muscle health through a nutritional lens. Further, to look at the potential benefits and harms of some commonly proposed dietary supplements in non-athlete adults, while exploring the emerging role of the gut–muscle axis. Inflammation appears central to cellular events. Several supplements were identified that, alone or in combination, may help optimise muscle health, particularly when combined with exercise or where a deficit may exist. Although supportive evidence is emerging, real-world clinical benefits remain to be substantiated. Though dietary supplements are generally safe, their regulation is less stringent than for medicines. Adherence to recommended dosage, seeking medical advice regarding possible side effects/interactions, and obtaining supplies from reliable sources are recommended.

## 1. Introduction

The benefits of exercise are increasingly recognised. These include maintaining musculoskeletal function and controlling body weight; reducing the risk of chronic disease; enhancing cognitive function, mood, and sleep; and managing stress. Central to these positives are healthy muscles. Effective muscle function depends on a balance between anabolic and catabolic processes. The latter may be altered by nutritional status, age, inactivity, and pathophysiological changes seen in menopause and disease states. Barriers to activity include pain, tissue swelling and injury, both detectable musculoskeletal damage and microtrauma. The importance of inflammation is increasingly being realised in the pathophysiology of catabolism, cachexia, and sarcopenia.

Sport science is now central to achieving optimal fitness for athletes. One aspect of this is the recognition of the importance of appropriate nutrition to support training, competition, and recovery. This may involve the calculated intake of carbohydrates, proteins, and fats, along with vitamins and minerals. Here, the role of dietary supplements has emerged. Consuming additional protein products, for example, is popular for building muscle mass and improving performance [1]. Similarly, supplements offering anti-inflammatory and antioxidant properties may ameliorate muscle damage and its associated symptoms [2]. Recently, the landscape has changed as more is learned about the gut-muscle axis. The potential for ingesting probiotics (live micro-organisms) or prebiotics, sufficient to alter the gut microbiome, has generated scientific interest [3].

Questions remain about the general population, which is largely sedentary or only occasionally involved in sport. These groups are underrepresented in current nutritional research. Rather than supplementing to maximise athletic performance, what evidence exists of their efficacy in supporting muscle function and recovery in subgroups such as peri- and menopausal women, the ill and elderly? Since these groups experience the greatest muscle loss and functional decline, this evaluation is of scientific importance. Practically, this review hopes to inform clinicians, policymakers, and consumers to make informed, evidence-based choices in an expanding, largely unregulated supplement market.

Clearly, many individuals consider dietary supplementation to be of value: Sales of single and combination products continue to rise. The value of the global dietary supplements market was estimated to be worth nearly USD 152 billion in 2021 [4]. It is expected to expand at an annual rate of 8.9% until 2030, notably in North America [5].

Our aim in this narrative review is two-fold: First, to examine the pathophysiology of muscle function through a nutritional lens; and second, to look at the potential benefits and harms of some commonly proposed dietary supplements in non-athlete adults. From this, we hope to make some pragmatic suggestions in the face of an established marketplace where supplementation is the norm for many [6].

## 2. Methodology

This narrative review comprised two parts. The first explored pathophysiological factors which may influence musculoskeletal function, and the second looked at whether dietary supplementation could be beneficial, and if so, what are likely to be the most successful candidates.

This work involved a multiple-author search of articles in English in PubMed and Google Scholar between 1 April 2015 and 31 May 2025. This was combined with an examination of governmental and organisational publications, a manual search of source articles, alongside discussions with all authors. Initial search terms included physical activity, dietary, food and nutritional supplements/supplementation.

A more detailed literature search was conducted separately for each nutrient considered to be of potential value. Primary outcome measures included muscle function, muscle performance, muscle mass/strength, activities of daily living (ADLs), exercise-induced muscle damage (EIMD), and delayed-onset muscle soreness (DOMS). Non-English studies, those focused on children <18 years of age, or solely involving athletes, were excluded.

## 3. Background

### 3.1. Muscle Homeostasis

Healthy muscle performance supports ADL, such as mobility, posture, and breathing. As a highly plastic tissue, skeletal muscle can grow (hypertrophy) and decline (atrophy) in response to a range of stimuli [7]. To maintain muscle mass, the balance between protein synthesis (anabolism; Muscle Protein Synthesis (MPS)) and breakdown (Muscle Protein Breakdown (MPB)) or catabolism must be tightly regulated [8]. The rate of MPS is primarily driven by anabolic cues such as food consumption, dietary protein intake and physical activity. A key anabolic hormone is insulin, released by the pancreas in response to eating. In addition to glucose metabolism, which primarily takes place in skeletal muscle, one of its functions is to stimulate muscle hypertrophy mediated by the secretion of insulin growth factor 1 (IGF-1) and activation of the PI3K-Akt-mTOR pathway [9]. By comparison, catabolic stressors contributing to MBP include illness, physical inactivity, and inflammation. An imbalance, where MPB exceeds MPS, leads to skeletal muscle loss, for example, in catabolic states such as advanced cancer [10]. It follows that a dysregulation of skeletal muscle metabolism can strongly impact insulin sensitivity and thus, glucose homeostasis [11]. Nutrients are fundamental in these mechanisms, regulating muscle homeostasis [12].

### 3.2. Sarcopenia, Age-Related Decline, ‘Sarcobesity’ and Effects of Weight Loss Treatments

With age, muscles become less responsive to normal anabolic signals, for example, a lower sensitivity to insulin (anabolic resistance) [13]. This blunted response partly explains the progressive decline in skeletal muscle mass (total amount of lean muscle tissue), strength (the amount of force generated by muscle contraction), physical performance (ability to carry out physical activities efficiently and ADL (basic activities of self-care, such as dressing, ambulation, or eating). observed in natural ageing. Gradual loss of muscle mass starts as early as age 30 and is estimated to increase by 1–2% each year from the age of 50 onwards [14]. Excessive muscle and soft tissue loss is termed sarcopenia.

Sarcopenia is a poorly understood multifactorial disease linked to chronic inflammation, nutrition, and physical inactivity [15]. Characterised by adverse muscle changes over time, the regulating factors linked to wasting are either activated (for example, members of the ubiquitin-proteasome system and myostatin) or growth factors are repressed (for example, IGF-1 and PGC-1α), resulting in loss of muscle protein and reduced function [9,16,17]. The same inflammatory state of sarcopenia is seen in ‘sarcobesity’ (SO), characterised by decreased muscle that coincides with increased fat mass. In recent years, the subset of individuals over 65 years of age with SO has increased considerably and is estimated to affect 2.1 billion by 2050 [18,19].

In overweight and obese individuals, sarcopenia progresses quickly when exercise is limited. Now a global health concern, both sarcopenia and SO are major contributors to frailty in the elderly [17,20,21]. With depleting muscle performance, ADL such as walking and climbing upstairs are compromised [22]. Physical inactivity becomes both a driver and a consequence of muscle loss. Impaired functional fitness reduces independence and quality of life, while the risk of falls, fractures and all-cause mortality increases [23].

A recently recognised cause of sarcopenia is significant iatrogenic weight loss. The advent of injectable therapeutic options such as liraglutide, semaglutide, and tirzepatide and bariatric surgery has shifted the treatment landscape for people who are overweight or obese. Crucially, while these interventions profoundly affect fat mass, they also impact muscle mass and bone density, particularly in older people more vulnerable to sarcopenia [24]. In a recent systematic review examining six studies of overweight or obese adults treated with the GLP-1 receptor agonist semaglutide, lean mass notably declined in individuals with total weight reductions ranging between 0 and 40% [25]. The loss of lean mass, that is, skeletal muscle, bone, and all other ‘fat-free mass’, has variable consequences for physiological health, including reductions in muscle strength and resting metabolic rate, alongside bone loss and an increased propensity for bone fracture [26]. Extrapolated, these have implications for aerobic capacity, weight maintenance and general quality of life. Currently, analysis of GLP-1 studies is limited by variability in study design and outcome measures.

### 3.3. Gender Differences in Muscle Loss

Sarcopenia can affect both sexes; however, women are more vulnerable. They may experience a worsening decline in muscle function, frailty and disability compared to men [27]. Several factors contribute to this process [28], including women generally having lower muscle mass than men, reduced oestrogen levels during peri- and post-menopause, decreased appetite levels, less physical activity, but higher degrees of oxidative stress and inflammation [29,30].

Limited evidence suggests that women may be more susceptible to EIMD. This may be due to the physiological changes that occur during menopause. Hormones, notably oestrogen, are positively correlated with muscle mass maintenance, as well as levels of IGF-1, insulin, growth hormone (GH) and dehydroepiandrosterone (DHEA) [31]. By comparison, a decline in oestrogen has been linked to muscle loss with age [32,33]. In addition, lower oestrogen is linked to increased visceral fat, reduced bone mineral density, higher cardiovascular risk and overall reduced quality of life [34]. It is worth noting that women are historically underrepresented in scientific studies focused on muscle loss and function [35]. Only 8% of studies have been conducted exclusively on females when evaluating dietary supplements and exercise adaptations [36]. Future key research priorities should include, for example, stratified randomised trials and responder analyses by baseline deficiency.

### 3.4. Inflammation and Muscle Function

Inflammation has an important role in muscle homeostasis. Chronic systemic low-grade inflammation, occurring in the absence of infection, is a feature which becomes more common with age. This age-related increase in blood and tissue pro-inflammatory markers is termed ‘Inflammageing’ [37]. This process may contribute to a range of age-related pathologies and is a significant risk factor for morbidity and mortality. Inflammageing is thought to result from a loss of control over systemic inflammation, leading to chronic overstimulation of the innate immune system. Inflammation markers such as tumour necrosis factor-α (TNF-α), C-reactive protein (CRP), and interleukin 6 (IL-6) are deleterious to skeletal muscle anabolism by triggering the pro-inflammatory NF-κB pathway (Figure 1). The latter promotes MPB via the ubiquitin-proteasome pathway, inducing inflammation and blocking muscle fibre regeneration [38]. Elevated levels of these inflammatory markers are not only found in sarcopenia but also in EIMD [13].

EIMD is caused by mechanical stress from unaccustomed, long-duration or strenuous high-intensity exercise. This, in turn, results in a pronounced inflammatory response, notably cytokine and reactive oxygen species (ROS) production, leading to decreased muscle function [2,39]. EIMD can affect non-athletes and individuals exercising as part of their treatment for sarcopenia. Features of EIMD include pain, delayed-onset muscle soreness (DOMS), reduced training quality, impaired performance, and interference with ADL. Tools to evaluate EIMD may include biochemical markers, such as creatinine kinase, subjective pain scales, functional tests and, in some cases, direct analysis of muscle biopsies.

In diseased or sedentary individuals, the presence of EIMD and DOMS may be more pronounced, even with limited activity [40]. Among overweight individuals, participation in high-intensity exercise, relative to moderate-intensity, was associated with lower levels of self-efficacy and in-task enjoyment [41]. These findings suggest that the increased physiological demands and recovery requirements of high-intensity activity may amplify certain psychological barriers to exercise. Although the inflammatory response to EIMD is thought to be integral to the muscle repair process [42,43], it may contribute to secondary muscle damage due to excessive macrophage accumulation, muscle swelling and delayed healing [44]. Chronic inflammation, in turn, may affect MPS and hamper the positive adaptations typically enhanced by training [45].

Gut health appears to be important in ‘inflammageing’ (vide infra). Gut microbiota dysbiosis is a term used to describe a reduction in microbial diversity. It is characterised by an imbalance between beneficial anti-inflammatory and harmful pro-inflammatory microbiotas [46,47]. This has downstream consequences for systemic inflammation, nutrient absorption, and muscle metabolism [48]. Although these changes vary between individuals [49], an unhealthy gut microbiota is associated with accelerated functional decline in older persons [50]. Since inflammation has been proposed as a contributing factor to anabolic resistance, this may, in turn, be affected by the gut microbiome. Thus, a link between gut and muscle health seems plausible [13].

### 3.5. The Gut–Muscle Axis

The human microbiota comprises 100 trillion bacteria with a total genome that is 150-fold larger than the human genome [51]. This complex ecosystem of microorganisms plays an important role in human health, influencing functions such as vitamin uptake, digestion and immunomodulation [52]. Both composition and diversity may affect skeletal muscle metabolism and function during anabolic (exercise or in athletes) and catabolic (sarcopenia and cachexia) phases [53].

Although physiologically distinct, skeletal muscle and the gut influence each other bidirectionally. The term gut-muscle axis is used to describe this association. For example, activity in the lumen and gut mucosa results in microbial metabolites, gut peptides, lipopolysaccharides, and interleukins, which in turn affect skeletal muscle function by regulating systemic/tissue inflammation and insulin sensitivity [53]. Imbalance in the microbial community (gut dysbiosis) is associated with an increase in Gram-negative bacteria that have endotoxic properties and upregulate pro-inflammatory cytokines IL-6, IL-1, TNF-α, and interferon gamma secretion, therefore inducing inflammation [54,55].

Conversely, skeletal muscle can affect the biodiversity of the gut microbiota. Muscular atrophy correlates with a decrease in the number of species sending anti-inflammatory and pro-anabolic mediators. Sarcopenia-associated intestinal dysbiosis and systemic weakness among the elderly have been linked to a number of pathophysiological changes. These include a reduction in muscle capillaries, increased intestinal barrier permeability, elevated blood lipopolysaccharide (LPS) levels, immune system activation, a decrease in insulin sensitivity, and more severe inflammation. These disruptions contribute to reduced mitochondrial biogenesis and function, as well as protein synthesis [13,56]. Taken together, there is a clear association between the gut microbiota and the regulation of skeletal muscle mass and function, as well as the development and worsening of metabolic dysregulation seen in obesity and insulin resistance (Figure 2).

Evidence also suggests that the intestinal microbiome contributes to a reduction in oxidative stress and exercise-induced inflammation. Gut dysbiosis increases ROS generation and free radical macromolecule devastation that may reduce physiological adaptation while contributing to skeletal muscle atrophy [57,58]. In addition, the microbiota has a role in skeletal muscle function by improving the availability of energy substrates such as glucose [45]. Further, the microbiome impacts muscle protein synthesis and mitochondrial biogenesis and function, as well as muscle glycogen storage [45]. Consequently, by modulating the immune response, oxidative stress, metabolic processes, and nutrient bioavailability, the microbiota can affect training adaptation.

A recent systematic review of 20 studies identified a correlation between sarcopenic functional outcomes and gut microbiome diversity. However, the authors emphasised the need for more clearly defined subgroups to delineate interindividual differences between responders and non-responders, as well as for longitudinal studies to validate these findings. Despite variability in interventional methodologies, sarcopenic outcomes remained strongly associated with gut microbial diversity [59]. From this, it might reasonably be inferred that altering the composition of the microbiome could offer therapeutic benefits.

### 3.6. Exercise, Inflammation and Muscle Recovery

Muscle regeneration is the process of resolving inflammation and restoring strength after injury, an essential process for maintaining muscle mass and function. One key factor delaying regeneration is low-grade systemic inflammation and oxidative stress, often triggered by EIMD. This can lead to secondary tissue injury if not efficiently resolved [60,61]. DOMS is commonly assessed using a numerical pain rating scale (NPRS), which varies in onset but typically develops between 24 h and 7 days after exercise. Owing to its transient effects on muscle stiffness, sensitivity, range of motion, and strength, DOMS may also create psychological barriers to continued exercise for some individuals [41,62]. Avoiding delays in resolution is critical to preserving skeletal muscle function [16]. Moreover, modern lifestyles compound the issue via physical inactivity and overconsumption of energy-dense, nutrient-poor foods that contribute to increased metabolic disorders, higher morbidity and mortality [63].

Regular exercise, notably resistance exercise training (RET), remains a cornerstone of health in slowing muscle loss and mitigating cognitive decline [64]. Controlled by hormones and growth factors, RET accelerates skeletal muscle hypertrophy. Protein synthesis is induced via insulin-like growth factor-1 (IGF-1) and inhibited by myostatin [65]. Sports associated with strength and endurance are also associated with positive changes in the intestinal environment and gut microbiota, including increased gut microbiota diversity and an abundance of health-promoting bacteria [66]. Taken together with evidence that improvements in gut microbiota diversity are linked to improvements in systemic inflammation, it is possible that exercise and improvements in gut health could enhance muscle growth and recovery synergistically. There is a need to explore this further.

### 3.7. The Relationship Between Physical Activity, Dietary Behaviour and Supplementation

Growing evidence demonstrates a robust relationship between physical activity and dietary behaviour that is bidirectional, underscoring the importance of both for optimal health. Principal Component Analysis revealed an alignment of healthy dietary habits with increased physical activity that is mutually reinforcing, suggesting that integrated lifestyle interventions can enhance population health and quality of life [67]. Patterns of supplement use closely mirror regular engagement in physical activity, with individuals maintaining consistent activity levels over time more likely to use dietary supplements. Moreover, those motivated by athletic performance or competition are significantly more inclined toward supplementation [68].

Dietary choices and physical activity not only affect disease risk but also body composition. Evidence from comparative studies reveals that active individuals tend to select diets higher in fibre and lower in fat, with vegetarians typically demonstrating lower body mass and visceral adiposity than omnivores, even without the inclusion of sport. Flexitarian dietary patterns appear particularly advantageous for weight management among women, reinforcing the case for tailored nutritional recommendations [69]. Furthermore, adverse lifestyle conditions such as reduced activity, increased consumption of unhealthy foods, and psychological stressors were found to contribute significantly to weight gain during periods of environmental stress, as observed in the COVID-19 lockdown [70]. Collectively, these findings suggest that synergistic modifications in both diet and physical activity are pivotal for achieving favourable body composition and improved health, while supplementation practices are shaped by physical activity, consistency, and individual motivation.

Similarly, dietary supplementation and physical activity synergistically enhance each other’s effectiveness in supporting muscle function and overall physiological health. Supplements such as calcium, vitamin D, omega-3 fatty acids, and whey protein have demonstrated greater efficacy when combined with consistent exercise, contributing to improvements in muscle mass, bone density, and reduced risk of sarcopenia in middle-aged and older women [71]. Structured training regimens are necessary to maximise the anabolic potential of specific supplements, particularly proteins and select botanicals [71,72]. Importantly, regular physical activity not only supports supplement efficacy but also fosters a physiological demand for additional nutritional support, highlighting the need for individualised assessment and targeted integration of supplements within the context of an active lifestyle.

## 4. Supplementation Candidates

### 4.1. How Might Supplements Help Support Muscle Function?

A Western lifestyle can be harmful. Reduced intake of vegetables, fruits, nuts and whole grains combined with increased consumption of processed foods, refined sugars and saturated fat can result in a deficit in vitamins, minerals, fibre, and antioxidants [73,74].

Nutrition is integral to muscle metabolism [14]. Much of the evidence on dietary supplements as adjuncts to increased physical activity does not account for individuals’ nutrient deficiency status; thus, high-quality, long-term studies are required to confirm their benefits. Indeed, the combination of RET with nutritional intervention provides a better stimulus for maintaining muscle strength and mass than RET alone [75]. Protein and vitamins E and D are known to affect anabolic stimuli, promoting the synthesis of muscle proteins while protecting against oxidative damage and loss of muscle mass [76].

In addition to promoting muscle mass, dietary supplements are widely employed in nutritional strategies to improve and maintain performance in sports and exercise [77]. They may support rapid recovery between bouts of damaging exercise/physical activity by reducing inflammation and oxidative stress, thus alleviating DOMS and enabling continued exertion [78]. Recovery is a crucial aspect of training: enabling muscles to repair and rebuild, thereby increasing strength and endurance over time. Adequate recovery minimises the risk of muscular injury and overuse.

For injured individuals, good nutrition is important for efficient recovery [79]. Since muscle loss results from inflammatory responses, dietary supplements with anti-inflammatory and antioxidant properties have the potential to reduce muscle damage and DOMS while maintaining strength [2]. Common treatment of DOMS using non-steroidal anti-inflammatory drugs (NSAIDs) may not necessarily improve symptoms [80], and even exert negative effects on local homeostasis and the central nervous system [81]. This may be because NSAIDs block or impair the initial stage of healing by interfering with the normal inflammatory processes. Therefore, alternative approaches that include dietary supplements with anti-inflammatory properties may be preferable [40]. However, because these may also impact the gut microbiome [82], any resultant imbalance could potentially exert a positive or negative effect on muscle and other organs [83]. Moreover, as warned by the National Institutes of Health (NIH), excess intake of some nutrients can be toxic, as exemplified by inadvertent over-supplementation with iron from multiple sources such as breakfast cereals, beverages, and supplements. Thus, taking dietary supplements is sometimes deleterious [84].

Numerous potential dietary supplements are proposed to have beneficial effects on skeletal muscle function, performance, mass/strength, and recovery in relation to exercise and injury. Many of these are listed in Table 1, with some of the leading candidates now discussed alone, or in combination.

**Table 1 nutrients-17-03495-t001:** Dietary supplements proposed to have beneficial effects on skeletal muscle function, performance, mass/strength and recovery in relation to exercise and injury.

Supplement	Source(s)	Claim(s)	Comments
Boron	Kreider, 1999 [85]	No strong evidence of benefits	Review of multiple studies
β-alanine	Hoffman et al., 2012 [86]	Delays the onset of fatigue during high-intensity exercise; efficacy is enhanced with co-supplementation of creatine	Precursor to carnosine, enhances intramuscular H+ buffering capacity
β-hydroxy-β-methylbutyrate	Rathmacher et al., 2025 [87]	Enhances synthesis and reduces breakdown of muscle	Metabolite of leucine, the strongest evidence exists for resistance training
Caffeine	Bilondi et al., 2024 [88]	Significantly increases muscle strength and endurance	Strong evidence, meta-analysis of 9 meta-analyses. Further studies needed on the effect in women
Carnosine	Cesak et al., 2023 [89]	Prevents sarcopenia; preserves cognitive function	ß-alanine precursor is more bioavailable
Chromium	Kreider, 1999 [85]	No strong evidence of benefits	Review of multiple studies
Creatine	Antonio et al., 2025 [90] Antonio et al., 2021 [91]	Improves short-term athletic performance, may reduce muscle catabolism in males	Facilitates the production of cellular ATP. Best combined with resistance exercise
Glutamine	Master & Macedo, 2021 [92]	No strong evidence for improvement in athletic or immune system performance	Popular, despite a lack of evidence
L-arginine	Tapiero et al., 2002 [93]	Supports energy metabolism; some anti-ageing effects	Precursor to creatine
L-carnitine	Sawicka et al., 2020 [94]	Improved muscle mass, exercise tolerance and cognition in centenarians with sarcopenia; no effect in women (65+)	Negligible effects reported in most men (18+)
Leucine	Goes-Santos et al., 2024 [95]	Stimulates muscle protein synthesis, particularly in older adults with sarcopenia	Most effective in combination with other amino acids, higher doses are required in older adults
Magnesium	Kirkland et al., 2018 [96]; Veronese et al., 2014 [97]	Required for many bodily functions, including muscle contraction, associated with improved physical performance in older women	Evidence for supplementation over dietary intake is inconclusive
Methylsulfonyl-methane	Butawan et al., 2017 [98]	Anti-inflammatory and antioxidant, reduces exercise-induced soreness	Downregulates cytokine expression via inhibition of NF-κB
Omega-3	Nunes et al., 2025 [99]; Dam et al., 2025 [100]	No significant evidence for improvements in muscle mass, function, or size following supplementation and resistance training	May cause a small gain in muscle strength assessed via chair rise performance
Potassium	Youn et al., 2009 [101]; Vinceti et al., 2016 [102]	Maintains cellular homeostasis; reduces high blood pressure, stroke and cardiovascular disease risk	May have stronger benefits for men
Prebiotics	Davani-Davari et al., 2019 [103]	Protective effects on many body systems, for example, the gastrointestinal, immune, and cardiovascular systems	Feeds gut microbiota to produce SCFAs that enter the circulation
Probiotics	Sánchez et al., 2017 [104]	Improve immune/gut barrier function; modulate gut–brain axis; produce neurotransmitters	Improve diversity in the microbiome by (re)introducing healthy bacteria to the gut
Protein	Antonio et al., 2024 [105]	Essential for body functioning; improves muscle mass growth and repair	Best combined with resistance exercise
Turmeric	Maughan et al., 2018 [72]	Anti-inflammatory: reduces muscle soreness, improves training capacity/recovery	May act as a free radical scavenger
Taurine	Seidel et al., 2019 [106], Merckx & Paepe, 2022 [107]	Anti-inflammatory by increasing cytokines and suppressing NF-kB signalling; Antioxidant properties by preventing excessive ROS production; ensures proper functionality of skeletal muscle by modulating chloride and potassium ion channels, cellular action potential and preventing muscle depolarisation	Prevents muscle catabolism through modulation of various pathways, evidence from animal studies
Vanadyl sulphate	Kreider, 1999 [85]	No strong evidence of benefits	Review of multiple studies
Vitamin B12	Fernandes et al., 2024 [108]	Maintains brain and nervous system function; deficiency linked to osteoporosis	Deficiency common among vegetarians and vegans
Vitamin D	Piotrowska et al., 2016 [109]; Barengolts, 2013 [51] Bello et al., 2021 [110]	Regulates body functioning and global homeostasis; affects inflammation and gut microbiome, no evidence for an effect on muscle recovery after exercise	Most of global population is deficient

### 4.2. Protein

Protein is essential for many body functions and remains one of the most popular dietary supplements on the market, taken by recreational and professional athletes alike [92,111,112]. Their use is mainly targeted at promoting muscle growth and repair after exertion [76,112,113,114].

For sedentary individuals, the consensus recommendation for average daily intake is 0.8 g per kilogram of body weight (g/kg); however, subsequent meta-analysis and nitrogen balance studies report historical underestimation [115]. It has been suggested that some groups need more, for example, older adults may benefit from 1.2 g/kg/day, while some professional athletes require 2 g/kg/day [116]. There seems to be no major difference in protein requirement between women and men, except that women generally weigh less and so need to consume less total protein [76,95,111,112,113,114]. In hormone loss–associated sarcopenia, protein requirements for muscle growth and maintenance may be elevated in women during peri- and post-menopausal periods; however, high-quality evidence supporting this is currently lacking [117].

Many studies have found that high protein consumption enhances muscle protein synthesis, lean body mass, and recovery in highly active people, such as professional athletes [92,111,118,119]. It is a common misconception that, in contrast, sedentary people do not need to consume much protein. Several recent studies have, in fact, suggested that the recommended daily allowance is too low for people at all activity levels, and for optimal health, it should be at least 1 g/kg of body weight [105,115,120,121,122].

Older people need more protein as they naturally lose muscle mass when ageing, which can progress to sarcopenia. The combination of a high-protein diet and resistance exercise has been found to mitigate sarcopenia in middle-aged and older adults [76,95]. There is some evidence that protein supplementation may reduce the risk of fracture in post-menopausal women with osteoporosis [123]. Moreover, associations between protein supplementation, higher bone mineral density, slowed bone loss, and reduced hip fracture risk are evident [105]. Additional protein has other health benefits, too—a study of over 3500 nurses found those who had protein-rich diets were significantly more likely to meet the criteria of healthy ageing, for example, no impairment to physical function or memory, good mental health, and freedom from 11 major chronic illnesses, including diabetes and multiple sclerosis [124].

Proteins are not created equally and can vary considerably in terms of absorption kinetics and their influence on anabolic processes. Rapidly digested sources, such as whey, induce an immediate increase in circulating essential amino acids like leucine and are often favoured following post-exercise recovery. In contrast, casein is digested and absorbed more slowly, prolonging the anabolic response [125]. Plant-derived proteins also have relatively slower kinetics and frequently lack a complete profile of essential amino acids, which may diminish their anabolic effects. The isolation and purification of plant proteins can enhance their absorption kinetics and subsequent effects on muscle-protein synthesis [126].

As proteins are composed of chains of amino acids, it is possible to take isolated amino acid supplements to focus on specific goals. Some research has found that branched-chain amino acid supplements can mitigate delayed-onset muscle soreness in the days following exercise [127,128]. Another common amino acid supplement is β-alanine, a crucial component of carnosine synthesis—carnosine delays fatigue during intense exercise by increasing the capacity of skeletal muscle to handle acid build-up—this is pronounced in older adults, as carnosine concentrations naturally fall with age [86]. Furthermore, β-alanine has some anti-inflammatory and anti-degenerative disease effects [129]. Similarly, L-arginine plays roles in the urea cycle, the immune system, and the production of nitric oxide, which helps regulate blood pressure; it is also a precursor of creatine. Increased intake has been found to mitigate the symptoms of coronary heart disease and hypertension, and to have anti-ageing effects, including improving cognitive function in dementia patients [93,130].

### 4.3. Amino Acids: Creatine and Leucine

Creatine is an amino acid derivative (requiring glycine, L-arginine, and L-methionine) essential to energy metabolism, storage, and muscle contraction, as part of ATP resynthesis. It can be used to enhance short-term physical performance by increasing the rate of resynthesis and delaying fatigue [91,131]. Creatine is one of the most widely used supplements—a normal diet contains 1–2 g daily, but supplementation of an extra 0.1 g/kg daily can effectively increase bodily stores. Although a loading phase/maintenance phase protocol is common, there are no significant differences when compared to continuous daily intake. Creatine supplements have been extensively studied in athletes, among whom they are especially popular. Several other populations may also receive benefits [91,131,132].

In older adults, combinations of creatine supplements and resistance training have been found to increase muscle mass and strength, along with improvements in day-to-day activities. This pattern is also true for those with established sarcopenia. Studies report improvements in markers such as leg press strength in those following a regimen of training and creatine supplementation, though supplementation alone was ineffective [91,95].

Creatine supplementation in women is understudied, but there are several factors which could suggest it may be less effective than in men: women have significantly lower stored levels of creatine (~75% less), slightly higher intramuscular concentrations (~10% more), and lower reported dietary intake [132]. Given that the menstrual cycle and fluctuating hormones can affect creatine levels, and that for post-menopausal women, a decline in oestrogen leads to loss of muscle/bone mass and strength, it could be suggested that supplementation might be especially beneficial to women. Several studies have found that, when combined with resistance training, increased creatine intake can combat this loss [133].

As the main benefits of creatine supplements are related to physical performance, they are not commonly taken by sedentary people. As above, the evidence suggests that their use is more beneficial when combined with regular exercise. However, a few studies have found that some people do respond to increased creatine intake without exercise, resulting in improved muscle strength. This finding requires further research in the light of variable age, sex, diet, and baseline creatine stores among participants [90,105].

Leucine has been shown to play a positive role in sarcopenic outcomes. It is understood to directly modulate protein turnover via the rapamycin (mTOR) pathway [134]. Consumption of leucine reduces muscle loss in older adults, and in some, may enhance muscle mass. Several studies have tried to demonstrate the effects of leucine supplementation on MPS in patients with sarcopenia. Results suggest that ingestion of 10 g/day of leucine alongside a 12-week resistance programme improved walking velocity, but not muscle strength (*p* = 0.056) or the chair-stand test (*p* = 0.085) [135]. Without exercise, 6 g/day of leucine for 13 weeks failed to improve lean body mass compared to placebo, in a study by Martinez-Arnau et al. [136]. However, the latter reported that the intervention group had improved their walking speed, suggesting a positive effect on muscle function.

This topic was examined more recently in an umbrella review by Gielen et al., who concluded that leucine can significantly affect muscle mass in elderly people with sarcopenia [137]. The authors recommend protein supplementation, on top of resistance training, to increase muscle mass and strength, in particular for obese persons and for ≥24 weeks [137]. What does seem to be important in elderly subjects is to administer a dose high enough to elicit a sufficient anabolic response and promote p70S6K phosphorylation [138]. Due to anabolic resistance in older adults, approximately twice as much dietary leucine than young adults is required to achieve similar increases in MPS [139,140,141]. Also, other AAs usually need to be in combination with leucine to support effective MPS [142]. Females may have a lower response than males to leucine stimulation of MPS and, therefore, may require more leucine. Based on fat-free mass (FFM), females may have an 11% higher leucine requirement [143].

### 4.4. β-Hydroxy-β-Methylbutyrate (HMB)

A recent position statement by the International Society of Sports Medicine is generally supportive of the use of β-hydroxy-β-methylbutyrate (HMB) as a safe dietary supplement across a range of populations [87]. These included athletes, non-exercising and sedentary individuals, patients with age-associated sarcopenia, and the inactive due to illness or injury.

HMB is a metabolite of the amino acid leucine, and a natural product in humans and other animals. Oral consumption of two forms of HMB studied to date, calcium HMB (HMB-Ca) and a free acid form of HMB (HMB-FA), appear to be safe in humans up to at least one year with no negative effects on glucose tolerance and insulin sensitivity.

HMB’s primary mode of action, similarly to leucine, appears to be to enhance MPS and suppress MPB. Though its activation of mTORC1 is independent of the leucine-sensing pathway (Sestrin2-GATOR2 complex). HMB facilitates muscle growth/repair, reduces muscle damage after exercise and promotes recovery, especially when consumed around the time of activity. Benefits may be seen in increased lean and reduced fat mass across different training populations. Aerobic performance may also improve. The anti-inflammatory effects of HMB may contribute to reducing muscle damage and soreness [87]. HMB is a widely used supplement among sportspeople, and its benefits appear most pronounced in combination with resistance training and dietary control [87].

In non-exercise settings and among untrained individuals, especially older adults, HMB supplementation may be useful to improve muscle function, performance, mass/strength, and quality. The combination of HMB supplementation with exercise may be beneficial for the treatment of age-related sarcopenia, cachexia, and frailty. Though these may depend on Vitamin D status [144], suggestive of a delicate interplay between micronutrients.

### 4.5. L-Carnitine

The amino acid derivative L-Carnitine (LC) has an established role in cellular metabolism and energy production. In healthy individuals, 95–97% of all carnitines (either free or acylcarnitine) is stored in skeletal muscle and the heart, with a much lower concentration found in plasma [145].

As a substrate for the carnitine acyltransferase enzymes, LC is necessary for transporting fatty acids into mitochondria for energy production via β-oxidation [146]. Efficient β-oxidation of muscle fatty acids is inversely related to the production of reactive oxygen species (ROS) [147], that is, it decreases ROS production, thereby mitigating oxidative stress. ROS activates the ubiquitin-proteasome system and accelerates the degradation of muscle proteins [9], which leads to sarcopenia. Free-fatty acids can activate the inflammatory IKK/NF-kB pathway [11], leading to increased nitric oxide synthase (iNOS) production and insulin resistance [148]. LC also plays a role in preserving membrane integrity and stabilisation of the mitochondrial (CoASH)/acetyl-CoA ratio, thereby preventing the accumulation of acetyl-CoA that is correlated with insulin resistance. Skeletal muscle insulin resistance is a major characteristic of obesity and the development of type 2 diabetes [149].

A deficiency in LC, that is, when plasma levels are below 20 μmol/L (across all age groups), is associated with impaired fatty acid utilisation, altered glucose utilisation, and insulin insensitivity [145]. LC dysregulation is seen in diseases such as diabetes, malnutrition and ageing and can result in carnitine deficiency [150]. Diet is the main source of total body carnitine levels (75%), with only 25% being synthesised endogenously [151]. From this, it follows that ingesting carnitine and the use of supplementation could be beneficial [152]. In women, exogenous LC intake may be particularly important, since circulating levels of LC are lower than in men [153]. However, it does not follow that increasing plasma levels will necessarily increase intramuscular LC [154].

To increase the total carnitine (TC) content in muscles, LC should be co-ingested with carbohydrates to induce an insulin response (insulin-mediated stimulation of muscle carnitine transport) [155]. Increasing muscle TC content in resting healthy humans in this combination approach reduces muscle glycolysis, increases glycogen storage and is accompanied by an apparent increase in fat oxidation [155]. This is because muscle-free carnitine availability may limit fat oxidation during high-intensity submaximal exercise [156].

Since LC is involved in maintaining skeletal muscle protein balance by means of proteolysis and protein synthesis [157], supplementation has the potential to support muscle function. Also, by mitigating oxidative stress by means of its antioxidant properties, supplementation with LC may be beneficial for the prevention and treatment of muscle damage and facilitation of muscle recovery [158]. LC supplementation has been shown to ameliorate muscle damage in both resistance-trained and untrained populations. Results indicate that in some cases, LC supplementation improves muscle soreness and markers of muscle damage (Creatine Kinase: CK, Lactate Dehydrogenase: LDH, and Myoglobin: Mb) [159]. Also, LC can enhance overall performance, measured by maximal and peak power, and maximal oxygen consumption (VO_2_ max), via reducing lactate production and increasing plasma glucose [160]. It is therefore not surprising that LC is widely used as a supplement among amateurs and athletes [94].

Orally administered LC can increase protein synthesis [161], by their metabolization into circulating trimethylamine-N-oxide (TMAO) [151]. This, in turn, is taken up by skeletal muscle [162]. Prolonged LC supplementation elevates circulating TMAO levels in healthy aged women [163]. Although supplemental consumption of carnitine is generally tolerated since it is easily eliminated [150], the use of uncontrolled amounts of carnitine supplements must be guarded against, as TMAO has been recognised as a novel risk factor for cardiovascular disease [164].

An overabundance of TMAO may be offset by prebiotics, nondigestible carbohydrates that encourage beneficial bacteria, and probiotics, live microorganisms which provide health advantages. Certain probiotic strains, such as *Lactobacillus plantarum*, *Lactobacillus reuteri*, and *Bifidobacterium longum*, have been shown to increase beneficial microbial populations while suppressing TMA-producing bacteria, thereby lowering TMAO levels. Prebiotics such as fructooligosaccharides and inulin promote the growth of bacteria that do not produce TMA. These dietary supplements aid in restoring gut microbial balance and reducing the hazards linked with TMAO [157].

### 4.6. Vitamin D

Vitamin D plays an important, pleiotropic role in maintaining global homeostasis. As well as being essential in regulating calcium and phosphorus balance, and assuring optimal functioning of major organs such as the skin [109], recent research highlights its direct role in muscle physiology function, with the identification of vitamin D receptors (VDR) on muscle cells [165]. Importantly, skeletal muscle serves as a reservoir for vitamin D, notably during the winter months [166], demonstrating the bidirectional link between the two.

VDR-associated molecular signalling pathways are known to contribute to the development and progression of inflammatory bowel disease (IBD) via their effects on gut barrier function, the microbial community, and the immune system. Vitamin D is an immunomodulator of the innate immune system, exerting anti-inflammatory effects and critically contributing to the maintenance of gut barrier integrity and modulation of the gut microbiota. These mechanisms may influence the development and progression of IBD [167]. Vitamin D intimately regulates inflammation through its bidirectional relationship with gut microbiota [51]. By partly controlling the gut microbial composition, the amount of dietary vitamin D and its circulating levels may maintain immune homeostasis in healthy individuals [168]. The NHANES III survey in the US showed that older adults with higher serum 25-hydroxyvitamin D [166] (vitamin D) levels (>94 nmol/L) performed better in physical function tests such as the 8-foot walk, and chair stand, compared to those with lower levels (<60 nmol/L) [169]. Additionally, grip strength has been found to correlate with circulating serum 25(OH)D levels [170].

Despite evidence of its importance, over 77% of Americans are considered vitamin D insufficient [171], and unfortunately, it is common in postmenopausal women [172]. Vitamin D deficiency (<25 nmol/L) is associated with decreased muscle size, in particular the large extensor and flexor muscles of the lower limbs, impairing muscle strength and functional mobility [173,174]. This reduced physical function, coupled with increased disability and frailty in older adults, leads to an increased risk of sarcopenia and falls [174,175]. Lombardi et al. report a possible role for vitamin D in managing pain associated with Fibromyalgia Syndrome, and for individuals suffering from chronic widespread musculoskeletal pain [176].

Evidence that vitamin D deficiency causes microbial imbalance in the gastrointestinal tract highlights it is an important modulator of the gut microbiota [167]. While sun exposure remains the most effective source, dietary intake is often insufficient due to limited dietary sources and poor absorption, making supplementation a promising means for achieving optimal serum levels [171]. There is evidence that vitamin D supplementation alleviates symptoms of deficiency, such as reducing unwanted changes in interleukin levels [173,177]. This confirms the anti-inflammatory effects of vitamin D, including for EIMD [177]. Indeed, supplementation of Vitamin D has been linked to improved athletic performance, particularly when initial vitamin D status is low [178].

### 4.7. Magnesium

Magnesium is essential for the regulation of skeletal and heart muscle contraction/relaxation, blood pressure, insulin metabolism, and it is required for the synthesis of DNA, RNA, and proteins [96]. Magnesium status has effects on muscle performance through its roles in energy metabolism and transmembrane transport. Approximately 27% of magnesium in the body is stored in skeletal muscle [179]. Magnesium deficiency is associated with poor physical performance [97], with low serum levels being linked to reduced muscle strength [180]. Similarly, a deficiency is associated with impaired intracellular calcium homeostasis, an increased inflammatory state, and muscle cell alterations attributable to increased oxidative stress [181]. Muscle mass and function are negatively impacted by these factors and could exacerbate the typical features of age-associated sarcopenia.

Supplementation with magnesium may be of benefit for skeletal muscle mitochondrial function. It may also be of benefit to prevent and/or treat sarcopenia. Along with other minerals such as iron, phosphorus and zinc, magnesium levels were found to be a positive predictor of change in limb lean mass, suggesting that greater intake could reduce muscle loss over time [182]. The evidence is, however, unclear. In healthy elderly women, daily magnesium oxide supplementation (for 12 weeks) generally improved physical performance [97]. Similarly, dietary magnesium could help conserve skeletal muscle mass and power in women of all ages [14].

By contrast, healthy young adults who were regular exercisers and free from hypomagnesemia experienced only modest ergolytic effects from short-term dietary magnesium chloride supplementation. Here, VO_2_ max, mean power output during a sprint test, and skeletal muscle mitochondrial respiration all decreased. The influence on performance during a 10 km time trial was negligible. This suggests that while magnesium supplementation could have a marginal role for some, the focus should be on eating high-quality, nutritious foods [183]. NB, different magnesium compounds may have different bioavailability [184].

### 4.8. Methylsulfonylmethane

The sulphur-containing molecule methylsulfonylmethane (MSM) has become a popular dietary supplement due to its anti-inflammatory and antioxidant properties [98,185]. The inhibitory effect of MSM on the pro-inflammatory signalling pathway NF-κB results in the downregulated expression of cytokines, including IL-1, IL-6, and TNF-α in vitro, with IL-1 and TNF-α being inhibited in a dose-dependent manner [186].

Because MSM has anti-inflammatory activities, it may be effective against activity-induced muscle soreness. This is generally caused by microtrauma to the muscles and surrounding connective tissue, promoting a local inflammatory response [98]. Research showed that MSM consumption for 28 days, or longer, reduced soreness in afflicted tissue in men completing a single bout of exercise, as well as in sedentary women and men with mild knee pain [187,188]. Endurance EIMD can be reduced with MSM supplementation, as assessed by measuring creatine kinase [189]. When tested clinically, pre-treatment with MSM reduced muscle soreness following both endurance [187], and strenuous resistance exercise [190]. Suppressing the expression of cytokines such as TNF-α [186], by MSM may reduce stimulated mitochondrial-generated ROS [191], confirming its antioxidant effect.

### 4.9. Potassium

Potassium (K^+^) plays a crucial role in maintaining normal membrane potential and cellular function, particularly in muscle and nerve cells. A continuous balance between dietary intake and excretion of K^+^ maintains total body K^+^ content. Critical to the acute regulation of extracellular fluid (ECF), K^+^ in skeletal muscle is buffered by shifting between intracellular (ICF) and ECF compartments primarily via the Na^+^, K^+^ ATPase pump [101,192].

The transition towards Westernised diets has led to a substantial decline in K^+^ intake compared with traditional food habits. Presently, a large proportion of the global population may now be experiencing suboptimal K^+^ intake [193]. Consequently, the World Health Organisation (WHO) recommends an increase in potassium intake to minimise the risk of cardiovascular disease and stroke in adults [102,194] and to reduce blood pressure [195]. A high potassium/low sodium diet is associated with better blood pressure control [196]. By comparison, there is an association between suboptimal dietary potassium intake, high blood pressure, cardiovascular diseases (for example, stroke and coronary heart disease) and kidney disease (chronic renal failure and kidney stones) [197].

Dietary potassium intake may also have a beneficial effect on bone health and insulin sensitivity [198,199], both of which are important for skeletal muscle anabolism. High potassium intake is associated with a reduced risk of insulin resistance and diabetes [200], while low intake is linked with lower insulin sensitivity [201]. Insulin resistance, in turn, is related to muscle catabolism in sarcopenia [202]. Further, impaired insulin signalling in cultured muscle cells stimulated proteolysis to cause atrophy [203]. Research in animal models demonstrated that a diet low in potassium promoted inflammation and was associated with increased free radical generation [204]. As discussed, chronic inflammation is well-known to be a risk factor for muscle mass loss [30]. Taken together, the close association between potassium intake and muscle mass may be orchestrated by their effects on insulin sensitivity and inflammation. High potassium intake may help reduce muscle protein breakdown by improving insulin resistance and lowering the inflammatory state.

Given that muscle mass is the primary store of potassium in the body, sarcopenic individuals typically present with a reduced body potassium content [205]. Therefore, potassium supplementation may be beneficial not only as a preventative measure but also for individuals with established muscle loss. Potassium also plays a critical role in optimising muscle contraction, fatigue resistance and adaptation to hormone replacement therapy in older women. In addition, ageing is associated with progressive difficulties in ion regulation, another factor that may impair muscle contraction [206].

There is growing evidence that potassium alkali intake is linked to favourable effects on muscle function, overall muscle health, and, potentially, the prevention of falls. These benefits may be due to potassium consumed as alkaline salts (for example, from fruits and vegetables) having a neutralising effect on acidity [193]. The acid-base theory suggests this might protect against loss of muscle mass [207]. In normal adults, metabolic acidosis stimulates nitrogen loss due to accelerated degradation of muscle protein [208]. Treatment with potassium bicarbonate has been shown to improve the nitrogen balance in postmenopausal women with mild metabolic acidosis [209].

Interestingly, gender may determine the association between potassium intake and muscle mass. Higher dietary potassium intake decreased the occurrence of low muscle mass in men but not in women. In men, dietary potassium intake may affect skeletal muscle mass independently of total energy intake, but in women, muscle mass may be affected by total energy intake rather than potassium intake [210]. Gender-specific differences in muscle protein metabolism, which are thought primarily to be associated with hormone profiles, may consequently lead to differences in therapeutic effects [211], as well as the fact that insulin resistance is higher in men than in women. This may be due to differences in adipokines and sex hormones [212]. Therefore, given its positive effects on insulin sensitivity, high potassium intake might be more beneficial for men than women.

### 4.10. Turmeric

Curcumin, a phenol compound found in the spice turmeric, has been shown to positively influence multiple pathways, including inflammation and protein degradation [213], thereby improving training capacity, muscle soreness, injury management and recovery [72].

The anti-inflammatory actions of curcumin may be due to the suppression of the NF-kB signal pathway, which affects IL-6, IL-1, and TNF-α, thus decreasing the expression of pro-inflammatory genes [214]. By limiting post-exercise inflammation, curcumin reduces pain sensitivity and DOMS. Further, the compound has been found to reduce muscle damage by decreasing creatine kinase (CK) [215] and other biological indices of muscle inflammation following EIMD [40]. Curcumin administration before, during, and up to 72 h after exercise has been shown to improve performance by lessening EIMD and modulating activity-induced inflammation [216]. Following 7 days of supplementation, significant improvements in muscle soreness, muscle damage, EIMD, and muscle strength have been demonstrated, as well as greater joint flexibility [217]. Importantly, curcumin is tolerable with no significant side effects in humans [216].

Research suggests that curcumin reduces secondary muscle damage by acting as a free radical scavenger to support muscle regeneration [218]. It also demonstrates direct antioxidant effects [216]. Curcumin has an ameliorating effect on acute and chronic pain pathways, including inhibition of oxidative stress and cyclooxygenase-2, and inhibition of apoptosis, among others [219].

### 4.11. Caffeine

Caffeine is a stimulant consumed by approximately 80% of people worldwide [220]. During the day, adenosine triphosphate (ATP) is degraded to adenosine, which then accumulates in the extracellular space [221]. The binding of adenosine to adenosine receptors in the brain causes tiredness. Caffeine, a type of receptor antagonist, is structurally similar and competes with adenosine for its receptor. By preventing adenosine binding, notably to adenosine receptor subtypes A_1_ and A_2_, alertness is enhanced [222].

Caffeine can also have ergogenic effects [220], increasing muscular endurance and enabling longer workouts. According to a meta-analysis of meta-analyses, caffeine significantly increases muscular strength muscle strength (SMD = 0.18, 95% CI: 0.14, 0.21; *p* < 0.001) and muscle endurance (SMD = 0.30, 95% CI 0.21, 0.38; *p* < 0.001) [88]. Further, a literature review has shown that moderate coffee drinking positively impacts the microbiome by increasing diversity, and through increasing the relative abundance of bacterial phyla, including Actinobacteria [223].

Unfortunately, the performance benefits of caffeine depend on the initial state of arousal, demonstrating a ‘U’-shaped dose effect. If this increases beyond a certain threshold, then performance decreases [224]. Further, caffeine is associated with generally mild adverse effects, limiting its general use as a supplement. These include anxiety, restlessness, fidgeting, insomnia, facial flushing, increased urination, irritability, muscle twitches or tremors, agitation, tachycardia or irregular heart rate, and gastrointestinal irritation. Variable effects of caffeine may have a genetic basis in some individuals, owing to polymorphisms in the adenosine receptor [225].

### 4.12. Omega-3 Polyunsaturated Fatty Acids

Omega-3 polyunsaturated fatty acids (*n*-3 PUFAs), especially eicosapentaenoic acid (EPA) and docosahexaenoic acid (DHA), have well-known anti-inflammatory and immunomodulatory properties. Their potential role in muscle metabolism is gaining attention. Proposed mechanisms for their anabolic or anti-catabolic effects include reducing chronic inflammation [226], enhancing MPS through mTOR (mammalian target of rapamycin) signalling [227], and improving neuromuscular function via changes in membrane fluidity and motor neuron preservation [228]. While these mechanisms are biologically plausible, the highest level of clinical evidence up to date, derived from an overview of 33 systematic reviews and meta-analyses by Nunes et al. [99], and a meta-analysis by Dame et al. [100],—concluded that omega-3 supplementation does not significantly improve muscle mass, function, or size, albeit there may be small, yet statistically significant gains in muscle strength.

The clinical impact of *n*-3 PUFAs appears context-dependent. In individuals performing resistance training, *n*-3 PUFAs may enhance lower-body strength, although they do not consistently boost muscle hypertrophy [99,100]. A meta-analysis of resistance training and *n*-3 PUFA interventions reported significant strength improvements but null effects on lean mass accretion [229]. In contrast, in physically inactive adults, effects are minimal: a meta-analysis [230] reported small improvements in strength but no significant changes in mass or function. Sex-based differences remain unclear due to underpowered or non-stratified studies, although some randomised controlled trials [231] suggest older women may experience greater benefits than men following omega-3 supplementation combined with resistance training. Among sarcopenic or frail older adults, some functional improvements, such as faster chair-rise times, have been observed, albeit increases in muscle mass or neuromuscular activation are not consistently reported [100]. For non-sarcopenic populations, evidence is weaker, with no meaningful effect on MPS, and only modest effects on whole-body protein synthesis [232]. Therefore, despite biological plausibility, current evidence does not support the routine clinical use of omega-3 supplementation to improve muscle health outcomes.

### 4.13. Prebiotics and Probiotics

The gut microbiome plays an important role in regulating muscle mass and function, modulating health, performance, and recovery after exercise. *Lactobacilli* and *Bifidobacterium* families are linked to a reduction in inflammation by affecting the secretion of anti-inflammatory cytokines. Relevant cytokines include IL-10, tryptophan-2,3-Dioxygenase (IDO) and TGF-β, which causes Treg stimulation (regulatory T cells), as well as Th1, Th2, and the inhibition of helper lymphocytes Th17 [233]. Specifically, *Bacteroides fragilis* can suppress the expansion of Th17 lymphocytes by producing IL-10 [234].

Further, compromised immune tolerance results in gut dysbiosis that impairs the function of epithelial and intestinal barriers, further leading to imbalances in pro-Th17 and anti-inflammatory Treg lymphocytes. These changes have been linked to several metabolic, intestinal, and cardiovascular conditions [235]. Probiotics, defined by the WHO and the Food and Agriculture Organisation (FAO) as live microorganisms that, when administered in adequate amounts, confer a health benefit on the host [236,237]; have shown promising effects as supplements, or adjunct therapy, in mitigating the effects of diseases caused by gut dysbiosis [235].

Probiotic supplementation is an effective method of (re-)introducing healthy bacteria into the gut microbiome [238]. Probiotics exert beneficial effects on the host by improving immunomodulation, gut barrier function, and production of neurotransmitters, as well as by modulating cellular components of the gut–brain axis [104]. The *Lacticaseibacillus* strain *rhamnosus* (*Lacticaseibacillus rhamnosus* GG (LGG), previously named *Lactobacillus rhamnosus* GG) is the most extensively documented probiotic strain [239]. It has been shown to have a positive immunomodulatory effect [240]. *L. rhamnosus* has the potential to lessen sarcopenia by modulating the gut microbiome and metabolites and is therefore considered a promising dietary supplement [142]. Trials have demonstrated the clinical efficacy of LGG supplementation in adults experiencing various health issues, including acute diarrhoea, antibiotic-associated diarrhoea, travellers’ diarrhoea, pain-related functional gastrointestinal disorders, respiratory tract infections, nosocomial infections, atopic disease, and as a vaccine adjuvant [241]. Also, daily intake of *B. longum* BB536 may contribute to maintaining physical condition in healthy adults [242]. Interestingly, the response to probiotic supplementation has been reported to vary according to genetic and gender differences, with probiotic supplementation resulting in a larger reduction in pro-inflammatory gut microbes in women than in men [3].

The beneficial effects of probiotic supplementation on aerobic and anaerobic performance in active individuals have been reported. Microbiota modulation via the *Lactiplantibacillus* strain *plantarum* was associated with improved endurance running performance [243]. Running time to fatigue in the heat is also improved by 4 weeks of intake of a multi-strain probiotic [244]. In a mouse model, supplementation with the LGG has been shown to improve overall exercise performance and physical strength [245]. In humans, older adults gained muscular strength and enhanced endurance when taking a multi-strain probiotic mixture supplementation of *Lactobacilli* and *Bifidobacterium* [246]. These suggest that exercise may feel more comfortable in individuals taking specific genera, such as *Lactobacilli* and *Bifidobacterium*, as well as specific species, such as LGG. These are also understood to alleviate gastrointestinal symptoms and discomfort [247], which can be caused by a single bout of high-intensity or endurance exercise [248]. Faster post-workout recovery is also observed following the ingestion of *Bifidobacterium longum* 35624 and *Lactiplantibacillus plantarum* PS128, compared to placebo groups (*p* < 0.05) [249,250,251]. This modulation of the gut microbiota may exert a positive effect by improving inflammation and oxidative stress [243].

A prebiotic is “a selectively fermented ingredient that results in specific changes in the composition and/or activity of the gastrointestinal microbiota, thus conferring benefit(s) upon host health” [252]. Degradation of prebiotics by the gut microbiota [103], that is, consumption by probiotics, positively affects the composition of the intestinal microbiota and its metabolic activity. The consumption of fructooligosaccharides (FOSs), a recognised class of prebiotics, by probiotic bacteria can strategically modulate the intestinal microbiota, leading to increased production of short-chain fatty acids (SCFAs). This contributes to a decrease in the pH of the colonic lumen and has a protective effect on the intestinal barrier by inhibiting the growth of intestinal pathogens [253].

A randomised, double-blind, placebo-controlled study in adults investigated the dose–response relationship of three different FOS dose levels (2.5, 5, and 10 g/day) on gut microbiota. The researchers found that FOS consumption increased the relative abundance of *Lactobacillus* and *Bifidobacterium* [254]. Compared with a placebo (Maltodextrin), higher doses of FOS promoted the selective production of *Lactobacillus*. Consumption of prebiotics increases bacterial diversity, while withdrawal had the opposite effect, suggesting their effects are short-lived. Interestingly, prebiotics that decrease the *Firmicutes* to *Bacteroidetes* ratio reduced gut permeability, metabolic endotoxemia and inflammation [254]. Further to the FOS-mediated increased bacterial diversity, a positive impact of FOS on butyrate-producing bacteria was also observed, reinforcing the role of prebiotics in conferring beneficial functions to the host [254].

Data on clinical benefit is limited. For example, the dose of prebiotics required to produce a measurable clinical benefit in muscle is unknown. The variability in populations and difficulty in measuring muscle mass and function accurately and with high reproducibility limit the discovery of specific strains able to optimise muscle mass and function [53]. Research in rodents showing that the probiotics lactic acid bacteria and *bifidobacteria* can limit sarcopenia and cachexia while enhancing performance has not been reported in humans. A further consideration is that a dose sufficient to elicit changes within the gut microbiome may be insufficient to exert a measurable clinical effect [8].

Another approach is to combine a probiotic with a prebiotic. The result known as a synbiotic is defined as “a mixture comprising live microorganisms and substrate(s) selectively utilised by host microorganisms that confers a health benefit on the host” [252]. The defining feature is a synergy between the two components that improves the survival of probiotic microorganisms in the gastrointestinal tract [255]. Research suggests a superior efficacy on human health compared to that of a probiotic or prebiotic alone [256]. Importantly, these effects translate to muscle health; for example, prebiotics FOS and inulin, in addition to probiotics *Lactiplantibacillus* and *Bifidobacterium,* significantly improved hand grip strength and self-reported exhaustion level in older patients [257].

### 4.14. Multi-Use Components

Multi-ingredient supplements (MIS) may be superior to ingesting a single item to support muscle mass and strength with ageing [258]. For example, there is evidence that a mixture of β-alanine and creatine supplements is more effective than creatine by itself, including increased fat loss and muscle gains [86]. O’Bryan et al. claim that the theoretical benefits of combining creatine, vitamin D, leucine, glutamine, and others to optimise function justify their use more broadly [259]. In combination, these components may act symbiotically or synergistically, producing an additive effect. For instance, leucine is associated with insulin levels [260], with insulin being required for the transport of LC into muscles, where it is known to exert a positive effect [154]. In support of this, a recent study reported increases in muscle mass and strength in older adults consuming LC mixed with leucine, creatine, and vitamin D for 8 weeks [261]. Here, the primary composite outcome improved by 63.5 percentage points in the combination group vs. placebo (*p* = 0.013). In comparison, somewhat simplistically, the addition of potassium and magnesium may optimise muscle contraction [262].

Optimising muscle function with prebiotics in combination with supplements known to stimulate muscle anabolism (for example, proteins) appears to be an option at all ages and for all types of patients [53]. HMB co-administered with the probiotic *Bacillus coagulans* GBI-30 may enhance the effectiveness of HMB in muscle recovery [263]. Amino acid absorption may also be improved by administration of other probiotics, including *L. paracasei* LP-DG and *L. paracasei* LPC-S01, collectively suggesting that amino acids should be combined with probiotic supplementation [264].

Combinations of HMB with arginine, lysine and glutamine, or creatine and/or ATP may increase its positive effects on lean body mass (LBM) [87]. Adding vitamin D may improve strength and function, especially in those with insufficient vitamin D levels. As touched on above, an individual’s initial levels of vitamin D may influence the benefits of an MIS on muscle function, including those with HMB [144]. In healthy older adults with low vitamin D, the combination of HMB-Ca and vitamin D3 improves muscle strength and physical function, even without exercise [144]. Similarly, elderly women experienced increased muscle mass, strength and functionality after 12 weeks of supplementing daily with HMB-Ca, L-arginine, and L-lysine without exercise [265]. Importantly, these benefits were maintained in both elderly women and men after 12 months [266]. Though, despite increases in muscle mass, in vitamin D deficiency, strength gains were limited [265,267]. It is difficult to establish any synergistic benefits of HMB with vitamin D3, as individual comparative studies are currently lacking. Also, not all combinations prove beneficial; for example, HMB in combination with high whey protein does not always result in an enhanced effect [268]. This highlights the value of employing a considered, tailored approach to dietary supplementation, rather than simply combining all potentially beneficial supplements in one product (Figure 3).

As with individual supplementation, combining exercise with MIS can be beneficial. Compared to exercise alone, co-ingestion of HMB-Ca with amino acids (L-glutamine, L-arginine and taurine) significantly increased training-induced changes in LBM, muscle strength, and power over 12 weeks [269]. Further, in an untrained adult population, an MIS including HMB, vitamin D3, choline, and caffeine, alongside a training programme, increased total body strength in comparison to a control group [270]. From the available evidence, it seems likely that a combination of MIS with RET may improve strength more than some non-MIS alternatives, though not necessarily compared to certain nutrients alone, that is, protein [259]. This suggests that individual ingredients in an MIS may drive the benefits. Here, external factors may also play a role in efficacy. It is noteworthy that the magnitude of the effect of a multi-ingredient protein supplement was greatest in the untrained and elderly undertaking RET compared to trained individuals and in younger people [259].

## 5. Supplements and the Gut

As discussed above, the gut is important in human health. As well as the gut-muscle axis, the microbiome has a bidirectional link with micronutrients. On the one hand, the gut microbiome can regulate micronutrient levels and absorption rates, influencing the bioavailability of minerals (calcium, iron, zinc, magnesium) and vitamins (such as B, C, D, and K) [271]. On the other hand, these micronutrients, when taken in the form of dietary supplementation, can affect microbiome composition and functionality [272]. Given that a balanced composition of gut microbiota is needed for supplying and converting nutrients, it is necessary to consider how supplements might impact the gut and vice versa. The bioavailability of magnesium, for example, is increased by strains belonging to Lactobacillus spp. following the consumption of dairy and vegetables, while the former can also exert a positive effect on gut microbiome composition and metabolism of vitamins B1 and D in people with type 2 diabetes, metabolic syndrome and obesity [273,274,275,276].

In a recent study, a negligible effect was observed on the gut microbiota following 8 days of magnesium supplementation. However, a ~1.7 log-fold decrease in *Bifidobacteria* was noted. The latter is a beneficial member of the gut milieu [183]. By comparison, vitamin D supplementation increases the diversity of the gut microbiota and is associated with improved gut health [277]. Notably, vitamin D increases the relative abundance of *Bacteroidetes* and decreases *Firmicutes*. This is potentially beneficial, as a high ratio of *Firmicutes* to *Bacteroidetes* correlates with obesity and other diseases [278].

As well as increasing muscle mass and function, HMB increases gut levels of SCFAs, which are essential for maintaining gut barrier function and reducing ROS [279,280]. Omega-3 also increases SCFAs and modulates the gut microenvironment, favouring the abundance of anti-inflammatory *Bifidobacterium* and *Lactobacillus* [243,280]. *Bifidobacterium* may be the main bacterium that modulates the utilisation of omega-3. In turn, increasing *Bifidobacterium* using probiotics and prebiotics can increase circulating levels of omega-3 PUFA [243].

Curcumin is linked to an increase in beneficial bacterial species such as *Bifidobacteria, Lactobacilli*, and *Blautia* spp., and a reduction in pathogenic bacteria associated with inflammation and gut dysbiosis, such as *Ruminococcus* spp. [281]. *Blautia* spp. is known to metabolise curcumin [282], demonstrating that the interaction is bidirectional. Both the biotransformation of curcumin by gut microbiota and the regulation of intestinal microflora by curcumin can regulate curcumin activity [283]. It is noteworthy that the response to curcumin administration differs between individuals, with microbiota displaying significant variation over time. Thus, some people may experience an effect, while others do not [281].

The length of supplementation is a consideration when discussing the efficacy of a nutrient. For example, long-term protein supplementation may have a negative impact on gut microbiota, decreasing the presence of *Bifidobacterium* [83]. To date, no study has investigated the potential association between all dietary supplements, the gut microbiota composition, and clinical parameters of physical function.

## 6. Safety

Dietary supplements are products intended to supplement rather than replace a healthy diet. They are not medicines, yet they can exert a strong effect on the body. There is a paucity of systematic studies of adverse effects [284]. Their regulation differs from prescription and over-the-counter products. Dietary supplements can be brought to the market without the support of clinical trials [284]. There is no requirement for demonstrating the efficacy for any health condition; however, manufacturers are not allowed to claim that the supplement can be used for treating or preventing any particular disease [284].

Manufacturers and suppliers are responsible for the composition, purity, and strength of their products, and for demonstrating their safety. The label claims must be truthful and not misleading, and contain details of the recommended dose and all ingredients, including fillers, binders, and flavourings. Unfortunately, quality control and adequate labelling are sometimes lacking among the numerous products available globally [285,286]. Tests performed by independent third-party organisations are currently needed to mitigate concerns regarding product quality and verify the safety of nutritional supplements [287,288].

Third-party testing programmes may assist in regulating the purity and safety of supplements, which entails independent third-party organisations performing specific tests verifying the safety of nutritional supplements [288]. Examples of third-party testing organisations include the NSF Certified for Sport (National Sanitation Foundation International, Ann Arbor, MI, USA) and Informed Sport (LCG, Teddington, UK), which test every batch of products before market release and attests products do not contain substances banned by sports organisations, supplement contents match printed labels, and unsafe and doping-related contaminants are not detectable [289].

Side effects of dietary supplements are uncommon and generally mild, for example, gastrointestinal disturbances and some drug interactions [285] (Table 2). For example, excess iron causes nausea and vomiting and may damage the liver, while vitamin K can reduce the ability of warfarin to prevent blood clotting. Also, consuming too much vitamin A can cause headaches and liver damage, reduce bone strength, and result in birth defects [84].

Like many issues around supplementation, claims of benefit and concerns over safety warrant closer examination: it has been suggested that one of the most popular supplements for muscle growth, creatine, might result in kidney disease. A group of experts met to review the evidence and concluded that the product was generally safe and potentially beneficial across a range of subgroups [91]. Problems are more likely at high doses and when consumed over long periods. Also, because of extra ingredients added to certain foods, consumption may already be more than intended, and some supplements may contain potentially detrimental artificial colourings, flavourings and sugar [284].

Prescription medicines should not be stopped in favour of a dietary supplement. Also, side effects can occur when taken with other supplements. Seeking advice from healthcare professionals (HCPs) regarding benefits, side effects, and interactions seems sensible, notably for those with health issues, undertaking surgery, who are pregnant or breastfeeding, or who are planning a change in diet and lifestyle. Unfortunately, individuals may not have access to the expert advice provided by health professionals that athletes might [290].

**Table 2 nutrients-17-03495-t002:** Frequent interactions and contraindications of some nutritional supplements involved in muscle health/homeostasis.

Nutritional Supplement	Medication	Interaction/Contraindication
Magnesium	Bisphosphonates	Magnesium can decrease the absorption of medications used to treat osteoporosis. Separating consumption between oral bisphosphonates and magnesium-rich supplements, or medications by at least 2 h is recommended [291].
	Antibiotics	Magnesium can form insoluble complexes with tetracycline antibiotics, such as doxycycline, as well as quinolone antibiotics, such as ciprofloxacin. These antibiotics should be taken at least 2 h before, or 4–6 h after, a magnesium-containing supplement [292].
Vitamin D	Orlistat	Vitamin D absorption from food and supplements can be reduced by the weight-loss drug orlistat (Xenical and alli) and a reduced-fat diet, resulting in lower 25(OH)D levels [293].
	Steroids	Corticosteroid medications, such as prednisone, prescribed to reduce inflammation, can reduce calcium absorption and impair vitamin D metabolism [294].
Potassium	ACE inhibitors and angiotensin receptor blockers	Treatments for hypertension and type 2 diabetes, including Angiotensin Converting Enzyme (ACE) inhibitors and angiotensin receptor blockers (ARBs), reduce urinary potassium excretion, which can lead to hyperkalaemia [295,296].
Carnitine	Pivalate-conjugated antibiotics	Carnitine interacts with pivalate-conjugated antibiotics, such as pivampicillin, that are used to prevent urinary tract infections [297]. Chronic administration of these antibiotics can lead to carnitine depletion. However, although tissue carnitine levels in people who take these antibiotics may become low enough to limit fatty acid oxidation, no cases of illness due to carnitine deficiency in this population have been described [298].
Omega-3	Warfarin (Coumadin) and similar anticoagulants	Omega-3 has antiplatelet effects at high doses and might prolong clotting times when it is taken with warfarin [299]. However, the risk of clinically significant bleeding is not impacted or produced by omega-3s [300,301].

## 7. Discussion

This narrative review paints a mixed picture. Key issues to emerge are the importance of inflammation in muscle function and gut health, the role of the gut muscle axis, and how nutrition supplementation, particularly when combined with exercise, can influence cellular events. Supportive evidence is building across different populations, but a clear clinical benefit is difficult to prove outside of a laboratory setting. The information available, though sometimes inconsistent and of limited quality, should not be ignored, especially for at-risk sub-populations. We have demonstrated that physical activity and dietary habits, as well as physical activity and supplement use, share a bidirectional and mutually reinforcing relationship, with evidence showing that each behaviour can enhance the effects and uptake of the other [71,72]. What seems important is adopting a holistic approach, making informed choices, and undertaking well-conducted long-term studies among those who need them most.

Being inactive, a change in hormones as seen with menopause, immobility, chronic illness and ageing are all detrimental to muscle health. Increasing numbers worldwide are adopting a Western lifestyle. This is characterised by sedentary behaviour and a lack of physical activity. The associated diet, low in vitamins, minerals, and fibre, is linked to a chronic inflammatory state and an increased risk of diseases such as type 2 diabetes and obesity [302,303]. The same persistent inflammatory state is observed in sarcopenia. In this context, attempting non-drug approaches to counteract these trends could be a promising strategy [18,19].

Sarcopenia, associated with a measurable decline in muscle mass and function, is seen among 10% of the community, not all of whom might be considered elderly. By comparison, 40–50% of nursing home residents are affected. Sarcopenia is now widely considered a global pandemic [304]. The frequency and diverse nature of the problem suggest an urgent need to prevent or reverse these processes. Unfortunately, our knowledge as to why these changes take place remains incomplete.

Loss of muscle mass and function results in individual frailty, an increased risk of falls, functional decline, and, ultimately, impaired quality of life [174]. Recent findings have indicated that some patients who increase their physical activity may also be affected by sarcopenia [82]. Other pathophysiological processes to add to the list include SO, where lack of exercise leads to a cycle of non-muscle weight gain and immobility, while iatrogenic weight-loss causes both fat and muscle loss and bone resorption [305]. Here, the indication that approximately 45% of weight loss is attributable to lean mass has spurred efforts to develop new GLP-1 medications that include additional pharmacological components aimed at preserving lean muscle mass [306]. Current research suggests that combining resistance exercise with dietary supplementation, specifically with nutrients such as creatine, leucine, vitamin D, and omega-3 fatty acids, may serve as helpful adjuncts to GLP-1 drugs for preserving lean muscle mass [307].

Increasing protein intake is generally beneficial for muscle mass and function in the elderly. Also, adding essential amino acids, notably creatine and leucine and their derivatives, such as HMB, antioxidant supplements, polyunsaturated fatty acids, minerals, and vitamin D, all have demonstrable benefits [76,308]. For instance, low muscle mass and loss of strength are associated with low serum 25-hydroxyvitamin D [166,172]. Interestingly, muscles serve as a dynamic storage site for vitamin D and thus play a central role in the maintenance of circulating 25-hydroxy vitamin D levels during periods of low sun exposure, demonstrating the importance of ensuring adequate intake [166].

A multicomponent approach combining nutrition with exercise is considered more effective than nutrition intervention alone for improving frailty [309]. However, when assessing the combination of nutrition and exercise interventions in older women at risk of sarcopenia, micronutrients were revealed to optimise muscle strength compared to macronutrients [262]. The only dietary supplement recommended to enhance muscle health for women in peri-menopause and post-menopause is a daily 10 mcg, or 400 IU, vitamin D supplement. Additional supplements are only indicated when there is a clinical need [310].

Deficiencies in vitamin D and vitamin B12 seem to negatively impact muscle strength gains in both high-speed resistance training (H-RT) and multicomponent training (MT). Whereas deficient levels of vitamin E, potassium, iron and magnesium may reduce the efficacy of H-RT in the same population [262]. Minerals, such as magnesium and selenium, may be important nutrients to prevent and/or treat sarcopenia [308]. It is worth emphasising that a similar effect cannot be expected across all populations: some people may be responsive to dietary supplementation while others may not [281].

The popularity of many performance-enhancing supplements has increased among amateur athletes [311]. Some of these demonstrate benefits that could be helpful to those who are critically ill and may help to preserve and restore lean body mass and function. However, the uncontrolled use of these agents can pose health risks and is underreported in the general population. The safest and most effective agents in enhancing athletic performance are considered to be creatine, branched-chain amino acids, and HMB [312]. The topic is usefully reviewed by the National Institute of Health [313].

This review has focused on emerging knowledge concerning activities in the gut and their effects on skeletal muscle. Dietary supplementation has a profound impact on the gut microbiome, which has now been revealed as a key regulator of muscle mass and function [82]. Interventions positively targeting the gut–muscle axis may help to delay age-related muscle wasting and dysfunction, and reverse the sarcopenic phenotype [271,314] by attenuating changes in gut microbiota that accompany sarcopenia via pre- and probiotic supplementation [271]. In aged mice, *Bifidobacteria* and *Lactobacillus* probiotic supplements significantly enhanced muscle mass, strength, and endurance capability [13]. In participants following exercise programmes, targeting the gut microbiota to overcome anabolic resistance may maximise responses. For those who cannot carry out vigorous exercise programmes, the ability to influence skeletal muscle function via the gut microbiota offers an alternative solution. Taken together, it may be reasonably concluded that dietary supplementation can enhance the positive effects of physical interventions on muscle health [53].

There are many things we do not know. While there is evidence that the leading dietary supplement contenders can make a difference to measurable physiological parameters, we do not generally know whether they are effective in the absence of a deficit. Further, how long should they be taken for, when, for example, before, during or after activity, whether they work better in combination with other agents and exercise and who will benefit. It is likely that some individuals taking supplements will receive psychological benefits [64]. The placebo effect is well recognised across all areas of medicine [315]. This is not to dismiss their place, but rather a plea for targeted administration and quality research into their long-term benefits. It seems likely that they can make a valuable contribution to muscle mass and function, and the treatment and prevention of sarcopenia across various subgroups, but data is lacking at present to allow for definitive recommendations.

## 8. Conclusions

Dietary supplementation can optimise muscle function, performance, mass/strength and ADL, reduce loss and aid recovery in non-athletes. Only limited subgroups have been studied, notably the athletic population. Unfortunately, studies are lacking where they are needed: in peri- and menopausal women, the immobile, and the elderly, where sarcopenia is an increasing issue. Combining multi-nutrient supplementation with exercise to maximise the anabolic potential is considered important. However, both detecting a nutritional deficit at the cellular level and demonstrating a positive clinical effect of supplementation are difficult outside of a laboratory setting. For some, there may be a psychological benefit from dietary supplementation.

While favourable data (and any evidence of harm) from taking supplements are inconsistent and of variable quality, these should not be dismissed, as our knowledge of inflammation modification, the microbiome and gut muscle axis continues to evolve. The benefits of additional protein, vitamin D and the electrolytes potassium and magnesium seem widely accepted. Other supplements highlighted in this article all have their supporters: creatine, leucine, HMB, L-carnitine, turmeric, *n*-3 PUFAs and pre- and probiotics. Most are safe in recommended doses, though interactions can occur. For some, co-ingestion and timing may be important. As is so often the case in complex interrelated systems, more research is needed across the whole field of dietary supplementation.

If taking a supplement is considered potentially beneficial, to complement rather than replace exercise and a high-quality diet, then administration should ideally be under medical supervision. Where possible, this should follow a body composition and nutritional deficiency assessment and be tailored to the individual’s current lifestyle, health issues and medication. Unfortunately, most individuals will continue to consume dietary supplements without any involvement from an HCP. Because maintenance of skeletal muscle depends on a range of anabolic and catabolic factors, they should be encouraged to ingest a multicomponent product from an established manufacturer following the dosing recommendations.

## Figures and Tables

**Figure 1 nutrients-17-03495-f001:**
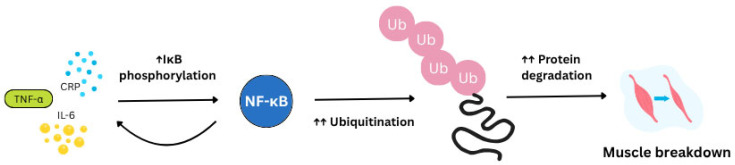
NF-kB Pathway Activation in Sarcopenia. Diagram representing the deleterious effect on muscle of elevated inflammatory markers, leading to activation of the inflammatory NF-kB pathway, and protein degradation via ubiquitination. Activation of the NF-kB pathway signals to increase inflammatory marker production, acting as a positive feedback loop to amplify the inflammatory response. Tumour necrosis factor-α (TNF-α); C-reactive protein (CRP); and interleukin 6 (IL-6).

**Figure 2 nutrients-17-03495-f002:**
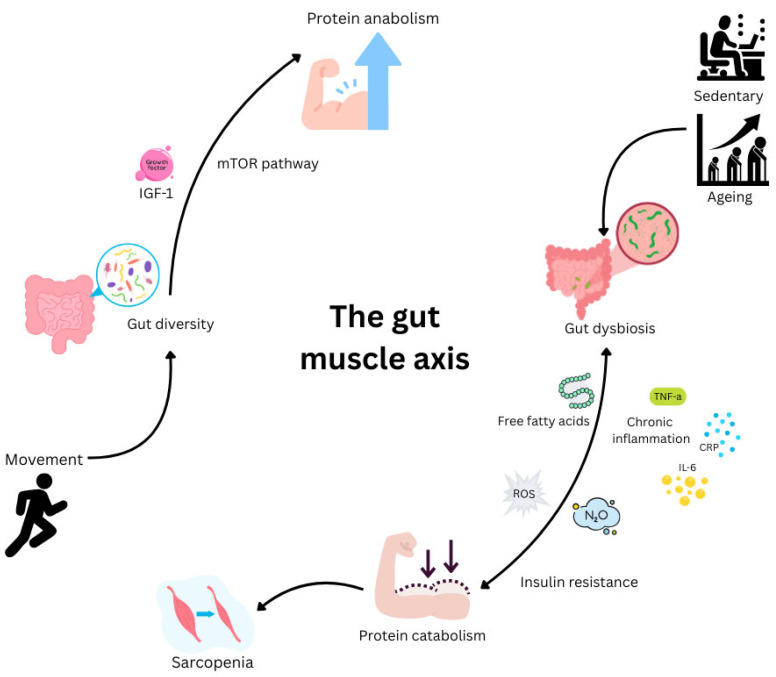
The gut muscle axis. Representative diagram of some of the pathways involved in muscle growth and muscle breakdown, and how gut health contributes to this. Being active improves gut microbiota diversity, catalysing insulin-like growth factor (IGF-1) production and the mTOR pathway, leading to muscle protein synthesis (MPS). An imbalance in the gut ecosystem (gut dysbiosis) is associated with muscle protein breakdown (MPB). Ageing and an inactive lifestyle decrease the diversity of the gut, triggering chronic inflammation, insulin resistance, MPB and ultimately sarcopenia.

**Figure 3 nutrients-17-03495-f003:**
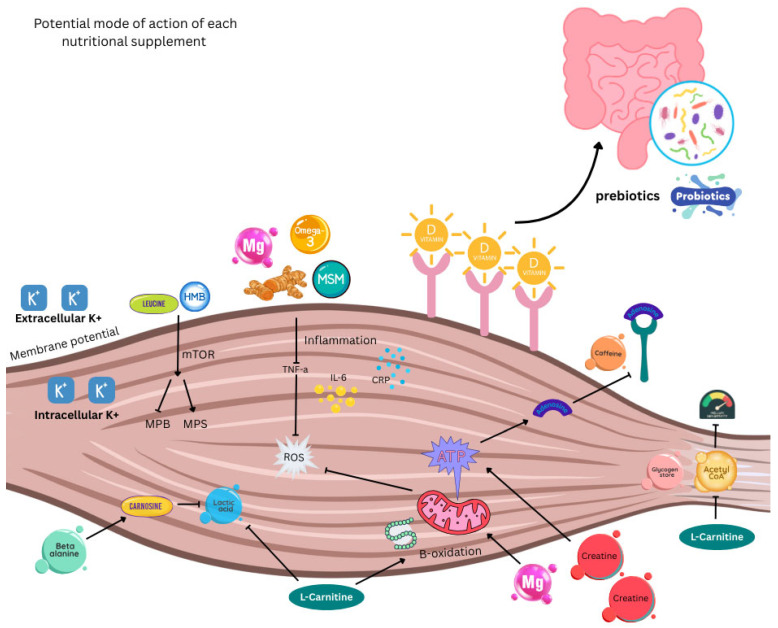
Representation of the theoretical mode of action of the dietary supplements mentioned in this review. ATP, adenosine triphosphate; CRP, C-reactive protein; HMB, β-hydroxy-β-methylbutyrate; IL-6, interleukin-6; K^+^, potassium; Mg, magnesium; MPB, muscle protein breakdown; MPS, muscle protein synthesis; MSM, methylsulfonylmethane; mTOR pathway, mechanistic target of rapamycin pathway; ROS, reactive oxygen species; TNF-α, Tumour Necrosis Factor Alpha.

## Data Availability

All data used in this narrative review is publicly available.

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
