# Peer review of "Can Dietary Supplements Support Muscle Function and Physical Activity? A Narrative Review"

_nutrients, 2025, doi:10.3390/nu17213495_

Round 1

Reviewer 1 Report

Comments and Suggestions for Authors

The manuscript is scientifically sound and well structured, with appropriate terminology and clear organization. The English is generally fluent and formal enough for a Nutrients submission. However, several issues related to conciseness, stylistic consistency, and phrasing could be improved to enhance readability and professionalism.

Abstract:

  • The abstract is coherent but could be made more direct:
    Replace phrases like “levels of exercise in the general population are mostly low” with “most adults engage in insufficient physical activity.”

  • The sentence “Supportive evidence is building but, outside of laboratory settings, clinical and real-world benefits require substantiation.” could be revised to “Although supportive evidence is emerging, real-world clinical benefits remain to be substantiated.”

  • Maintain uniform usage of key terms throughout:

    • muscle function → when referring to physiological or functional aspects

    • muscle performance → for athletic or exercise outcomes

    • muscle mass/strength → for sarcopenia or structural context

  • Several sentences are unnecessarily long or repetitive.
    Example: “This work involved a multiple author search of articles in English in PubMed and Google Scholar between 1st April 2015 and 31st May 2025…”

    change to:
    “Multiple authors independently searched PubMed and Google Scholar (April 2015–May 2025) for English-language studies…”

  • Consider breaking long paragraphs into shorter, more focused ones, particularly in sections 3.4–3.6 and 4.2–4.7.

  • Some sentences contain too many clauses, making them harder to follow.
    Example: “Although skeletal muscle tissue is physiologically distinct from the alimentary tract, both appear to influence each other bi-directionally.”

    change to
    “Although physiologically distinct, skeletal muscle and the gut influence each other bidirectionally.”

  • Check minor grammatical and stylistic issues.
  • Replace “i.e.” and “e.g.” with “that is” and “for example” in formal narrative contexts. Ensure space consistency after reference brackets (e.g., [14,15], not [14, 15] if following journal style)

Author Response

Nutrients-3948215

Review

Can dietary supplements support muscle function and physical activity? A narrative review

Authors: Louise Brough * , Gail A Rees * , Lylah Drummond-Clarke , Jennifer Elizabeth McCallum , Elisabeth Taylor , Oleksii Kozhevnikov , Steven Walker

Response to Reviewer 1

The manuscript is scientifically sound and well structured, with appropriate terminology and clear organization. The English is generally fluent and formal enough for a Nutrients submission. However, several issues related to conciseness, stylistic consistency, and phrasing could be improved to enhance readability and professionalism.

We appreciate the reviewer’s comments which have resulted in a number of beneficial changes. Thank you.

Abstract:

  • The abstract is coherent but could be made more direct:
    Replace phrases like “levels of exercise in the general population are mostly low” with “most adults engage in insufficient physical activity.”

Amended Page 2

  • The sentence “Supportive evidence is building but, outside of laboratory settings, clinical and real-world benefits require substantiation.” could be revised to “Although supportive evidence is emerging, real-world clinical benefits remain to be substantiated.”

Amended Page 2

  • Maintain uniform usage of key terms throughout:
    • muscle function → when referring to physiological or functional aspects
    • muscle performance → for athletic or exercise outcomes
    • muscle mass/strength → for sarcopenia or structural context

We have tried to keep to the suggested format throughout. A number of changes are highlighted throughout the manuscript.

  • Several sentences are unnecessarily long or repetitive.
    Example: “This work involved a multiple author search of articles in English in PubMed and Google Scholar between 1st April 2015 and 31st May 2025…”

change to:
“Multiple authors independently searched PubMed and Google Scholar (April 2015–May 2025) for English-language studies…”

We have revised this sentence Page 3. We have also made changes at other places throughout the manuscript.

  • Consider breaking long paragraphs into shorter, more focused ones, particularly in sections 3.4–3.6 and 4.2–4.7.

We have broken up several paragraphs e.g. Pages 6, 7 and 18.

  • Some sentences contain too many clauses, making them harder to follow.
    Example: “Although skeletal muscle tissue is physiologically distinct from the alimentary tract, both appear to influence each other bi-directionally.”

change to
“Although physiologically distinct, skeletal muscle and the gut influence each other bidirectionally.”

Change made Page 6

  • Check minor grammatical and stylistic issues.
  • Replace “i.e.” and “e.g.” with “that is” and “for example” in formal narrative contexts. Ensure space consistency after reference brackets (e.g., [14,15], not [14, 15] if following journal style)

Changes made throughout

Reviewer 2 Report

Comments and Suggestions for Authors

Can Dietary Supplements Support Muscle Function and Physical Activity? A Narrative Review

General comments

The paper addresses an applied and timely question: whether common dietary supplements can support muscle function and physical activity in non-athlete adults, focusing on inflammation, sarcopenia/sarcobesity, the gut–muscle axis, safety, and subgroups such as peri- and post-menopausal women. The narrative is wide-ranging and readable, and the “Safety” and “Women/EIMD” sections are useful to clinicians and educators. The manuscript would benefit from (i) clearer reporting of the literature search and study selection, (ii) explicit appraisal of evidence quality (e.g., a strength-of-evidence table or GRADE-style summary), (iii) greater consistency in terminology and scope, and (iv) tighter alignment between the article type (narrative review) and statements in the back matter (which currently say “scoping review”). The declarations are appropriately transparent but, given multiple industry affiliations, the funder’s role needs to be clarified beyond “arm’s length.”

Specific comments

Abstract

  1. The phrase “can optimise muscle function” would need a lighter tone to reflect the cautious tone used later in the piece too, e.g., “may help optimise… when combined with exercise or in deficiency contexts.”
  2. The Abstract also presents the population (non-athlete adults and specific subgroups) but not the principal functional outcomes (muscle strength, mass, ADL, or recovery from EIMD/DOMS). A short statement could be inserted stating which of the following outcomes are the main focus of the review, enhancing clarity and consistency with the Methods section.

Introduction

  1. One-sentence justification of key questions (population, types of supplements, and main outcomes) at the conclusion of the Introduction would better orient readers.
  2. "The global dietary supplements market was valued at almost USD 152 billion in 2021" is appropriately cited and provides the year. However, update this figure with a new estimate if possible as the pace of the market in this category has increased rapidly in recent years such as 2023–2025.

Methodology

  1. The search description lists databases, dates, and keywords but lacks detail. Please include: complete Boolean strings, whether MeSH terms or filters were used, screening and exclusion procedures, whether screening was done independently by two reviewers, and a concise PRISMA-style flow diagram.
  2. Define the primary outcomes in advance (e.g., muscle strength, mass, ADL, EIMD/DOMS indices) and clarify how mixed populations or heterogeneous dosages were handled.

Background

Muscle Homeostasis-

Very informative molecular mechanisms explained. Include 1 final sentence connecting these mechanisms (e.g., mTOR, inflammation, antioxidant signaling) to the supplements we discussed further.

Sarcopenia, Age-Related Decline, ‘Sarcobesity’ and Effects of Weight Loss Treatments-

  1. In Section 3.2, following the discussion of sarcopenia combined with the triad of muscle mass, muscle strength, and physical performance, it is important to insert a small clause that provides clear definitions of your outcome language in accordance with a recognized consensus and what domains your review highlights. This will be used to clarify how later outcomes and the tables relate to the diagnostic domains.
  2. The GLP-1–related weight loss should comment briefly on study design variability and the clinical meaning of lean-mass reductions.

Gender Differences in Muscle Loss-

This is an important point — women are under-represented. You could include a small summary box or table on sex-specific evidence and data gaps.

Inflammation and Muscle Function-

  1. The spelling of “inflammaging/inflammageing” must be consistent and defined once.
  2. Add a schematic figure connecting TNF-α/IL-6/CRP → NF-κB → UPS pathway to muscle functional outcomes; this would guide readers into the supplement section.

The Gut–Muscle Axis-

The bidirectional model and heterogeneity discussion are strong. End the section with a practical implication sentence (e.g., on responder vs. non-responder variability or intervention duration).

Exercise, Inflammation, and Muscle Recovery-

Specify common EIMD/DOMS measurement tools and time windows, and clarify the evidence strength for links between training intensity and psychological barriers.

Supplementation Candidates

How Might Supplements Help Support Muscle Function?-

Clearly separate mechanistic plausibility from functional clinical outcomes, and consider adding a boxed “Practical Note” on evaluating nutrient deficiency before supplementation.

Protein-

  1. The claim that ≥1 g/kg/day is required should include the level of evidence and specify applicability to non-athletic adults.
  2. Summarize differences among protein sources (whey, casein, plant) and absorption kinetics relative to muscle-protein synthesis.

Amino Acids: Creatine and Leucine-

  1. After noting limited female data, outline key research priorities (e.g., stratified randomized trials, responder analyses by baseline deficiency).
  2. Clarify leucine threshold effects in older adults and note that leucine must usually be combined with other essential amino acids.

Safety

  1. Good section; including a succinct table of frequent interactions/contraindications and a list of who should seek medical advice would be useful.
  2. Add a short note on third-party testing and variability of labels to address concerns about product quality.

Conclusion

The conclusion is well, cautiously so. Add one last sentence that supplements must complement rather than replace exercise and a high-quality diet, and that testing for deficiencies should be done before supplementation.

Author Response

Manuscript ID: nutrients-3948215

Type of manuscript: Narrative review

Title: Can dietary supplements support muscle function and physical activity?

A narrative review

Authors: Louise Brough *, Gail A Rees *, Lylah Drummond-Clarke, Jennifer Elizabeth McCallum, Elisabeth Taylor, Oleksii Kozhevnikov, Steven Walker

Response to Reviewer 2

(Prev. Reviewer 1)

General

 The paper addresses an applied and timely question: whether common dietary supplements can support muscle function and physical activity in non-athlete adults, focusing on inflammation, sarcopenia/sarcobesity, the gut–muscle axis, safety, and subgroups such as peri- and post-menopausal women. The narrative is wide-ranging and readable, and the “Safety” and “Women/EIMD” sections are useful to clinicians and educators.

The manuscript would benefit from (i) clearer reporting of the literature search and study selection, (ii) explicit appraisal of evidence quality (e.g., a strength-of-evidence table or GRADE-style summary), (iii) greater consistency in terminology and scope, and (iv) tighter alignment between the article type (narrative review) and statements in the back matter (which currently say “scoping review”). The declarations are appropriately transparent but, given multiple industry affiliations, the funder’s role needs to be clarified beyond “arm’s length.”

Comments

We are grateful to the reviewer for the positive comments and helpful suggestions. These have served to make a better paper.

(i-ii) This major narrative review (317 references) sought to obtain a broad overview of dietary supplementation. The reviewer is evidently an expert in their field. Therefore, we humbly suggest that a PRISMA diagram and formal evaluation of the evidence is not normally part of the process when developing a narrative review in the same way it is for a systematic review. Similarly, predefined inclusion criteria are not generally required. We accept that this flexible approach can make narrative reviews more susceptible to selection and interpretation bias compared to systematic reviews.

We hope that the reviewer will recognise that we have attempted to discuss numerous dietary supplements in a neutral manner. In controversial areas where evidence and opinions may differ, we have selected key references in conjunction with the lead author experts to provide balance. While we did not formally evaluate the quality of each paper in this huge work, most would probably be considered weak.

(iii) As suggested by (new) Reviewer 1, we have revised certain terminology throughout  e.g. ‘muscle function’, ‘muscle performance’ and ‘muscle mass/strength’.

(iv). In the Declaration section, we have clarified the funder’s role.

Abstract:

  1. The phrase “can optimise muscle function” would need a lighter tone to reflect the cautious tone used later in the piece too, e.g., “may help optimise… when combined with exercise or in deficiency contexts.”

Change Page 2:

Several supplements were identified which alone or in combination may help optimise muscle function,

  1. The Abstract also presents the population (non-athlete adults and specific subgroups) but not the principal functional outcomes (muscle strength, mass, ADL, or recovery from EIMD/DOMS). A short statement could be inserted stating which of the following outcomes are the main focus of the review, enhancing clarity and consistency with the Methods section.

Change Page 2:

By focusing on function, performance, mass and strength, ADL, exercise-induced muscle damage and delayed onset muscle soreness, this review sought to examine muscle health through a nutritional lens.  

Introduction:

  1. One-sentence justification of key questions (population, types of supplements, and main outcomes) at the conclusion of the Introduction would better orient readers.

On Page 3 Paragraph 2 we write:

“Questions remain regarding the general population who are largely sedentary or only occasionally involved in sport. Is dietary supplementation of value for improving muscle function and recovery in these populations?”

  1. "The global dietary supplements market was valued at almost USD 152 billion in 2021" is appropriately cited and provides the year. However, update this figure with a new estimate if possible as the pace of the market in this category has increased rapidly in recent years such as 2023–2025.

Change (Page 3):

We have added extra text and an additional reference (Zovi et al 2025)

The value of the global dietary supplements market was estimated to be worth nearly USD 152 billion in 2021[4]. It is expected to expand at an annual rate of 8.9% until 2030, notably in North America [5].

Methodology:

  1. The search description lists databases, dates, and keywords but lacks detail. Please include: complete Boolean strings, whether MeSH terms or filters were used, screening and exclusion procedures, whether screening was done independently by two reviewers, and a concise PRISMA-style flow diagram.

Comment:

As above, this is a scoping review and aims to explore broad themes, identifying literature gaps, mapping key concepts.

  1. Define the primary outcomes in advance (e.g., muscle strength, mass, ADL, EIMD/DOMS indices) and clarify how mixed populations or heterogeneous dosages were handled.

Change Page 4:

A more detailed literature search was conducted separately for each nutrient considered of potential value. Primary outcome measures included muscle function, muscle performance, muscle mass/strength, activities of daily living (ADL), exercise-induced muscle damage (EIMD) and delayed onset of muscle soreness (DOMS).

Background

Muscle Homeostasis-

Very informative molecular mechanisms explained. Include 1 final sentence connecting these mechanisms (e.g., mTOR, inflammation, antioxidant signaling) to the supplements we discussed further.

Change:

Added new sentence and reference (Page 4)

Nutrients are fundamental in these mechanisms, regulating muscle homeostasis (Adegoke et al., 2022).

Sarcopenia, Age-Related Decline, ‘Sarcobesity’ and Effects of Weight Loss Treatments-

  1. In Section 3.2, following the discussion of sarcopenia combined with the triad of muscle mass, muscle strength, and physical performance, it is important to insert a small clause that provides clear definitions of your outcome language in accordance with a recognized consensus and what domains your review highlights. This will be used to clarify how later outcomes and the tables relate to the diagnostic domains.

Change (Page 4)

This blunted response partly explains the progressive decline in skeletal muscle mass (total amount of lean muscle tissue), strength (the amount of force generated by muscle contraction), physical performance (ability to carry out physical activities efficiently and ADL (basic activities of self-care, such as dressing, ambulation, or eating). observed in natural ageing.

  1. The GLP-1–related weight loss should comment briefly on study design variability and the clinical meaning of lean-mass reductions.

Change (Page 5)

The loss of lean mass, that is skeletal muscle, bone and all other ‘fat-free mass’, has variable consequences for physiological health including reductions in muscle strength and resting metabolic rate alongside bone loss and an increased propensity for bone fracture [26]. Extrapolated, these have implications for aerobic capacity, weight maintenance and general quality of life.  Currently, analysis of GLP-1 studies is limited by variability in study design and outcome measures.

Gender Differences in Muscle Loss-

This is an important point — women are under-represented. You could include a small summary box or table on sex-specific evidence and data gaps.

Response:

Thank you for this suggestion. You will see that we have added new material including multiple references, a figure and table as suggested by you and the other reviewers and are reluctant to expand the manuscript further. The topic of gender differences could be an interesting future article.

We have added a further line:

Only 8% of studies have been conducted exclusively on females when evaluating dietary supplements and exercise adaptations[35].  Future key research priorities should include, for example, stratified randomised trials and responder analyses by baseline deficiency.

Inflammation and Muscle Function-

  1. The spelling of “inflammaging/inflammageing” must be consistent and defined once.

Changes made throughout

This age-related increase in blood and tissue pro-inflammatory markers is termed 'Inflammageing’ (Ferrucci et al., 2018).

  1. Add a schematic figure connecting TNF-α/IL-6/CRP → NF-κB → UPS pathway to muscle functional outcomes; this would guide readers into the supplement section.

A new figure is included: on Page 6

Figure 1. NF-?B Pathway Activation in Sarcopenia.

The Gut–Muscle Axis-

The bidirectional model and heterogeneity discussion are strong. End the section with a practical implication sentence (e.g., on responder vs. non-responder variability or intervention duration). -

Change: New text added (Page 7)

A recent systematic review of 20 studies identified a correlation between sarcopenic functional outcomes and gut microbiome diversity. However, the authors emphasised the need for better-defined subgroups to delineate interindividual differences between responders and non-responders, as well as for longitudinal studies to validate these findings. Despite variability in interventional methodologies, sarcopenic outcomes remained strongly associated with gut microbial diversity [56]

Exercise, Inflammation, and Muscle Recovery-

Specify common EIMD/DOMS measurement tools and time windows, and clarify the evidence strength for links between training intensity and psychological barriers.

Change Page 6

EIMD

Tools to evaluate EIMD may include biochemical markers, for example creatinine kinase, subjective pain scales, functional tests and in some cases, direct analysis of muscle biopsies

DOMS assessment is addressed on Pages 8-9

DOMS is commonly assessed using a numerical pain rating scale (NPRS), varies in onset but typically develops between 24 hours and 7 days after exercise. Owing to its transient effects on muscle stiffness, sensitivity, range of motion, and strength, DOMS may also create psychological barriers to continued exercise for some individuals [37, 59]  

Supplementation Candidates

How Might Supplements Help Support Muscle Function?-

Clearly separate mechanistic plausibility from functional clinical outcomes, and consider adding a boxed “Practical Note” on evaluating nutrient deficiency before supplementation.

Protein-

  1. The claim that ≥1 g/kg/day is required should include the level of evidence and specify applicability to non-athletic adults.

Change Page 13:

For sedentary individuals, the consensus recommendation for average daily intake is 0.8 grams per kilogram of body weight (g/kg), however, subsequent meta-analysis and nitrogen balance studies report historical underestimation [107].

  1. Summarize differences among protein sources (whey, casein, plant) and absorption kinetics relative to muscle-protein synthesis.

Change Page 13

Proteins are not created equally and can vary considerably in terms of absorption kinetics and their influence on anabolic processes. Rapidly digested sources, such as whey, induce an immediate increase in circulating essential amino acids like leucine and are often favoured following post-exercise recovery. In contrast, casein is digested and absorbed more slowly, prolonging the anabolic response [117]. Plant-derived proteins also have relatively slower kinetics and frequently lack a complete profile of essential amino acids, which may diminish their anabolic effects. The isolation and purification of plant proteins can enhance their absorption kinetics and subsequent effects on muscle-protein synthesis [118].

Amino Acids: Creatine and Leucine-

  1. After noting limited female data, outline key research priorities (e.g., stratified randomized trials, responder analyses by baseline deficiency).

Change: Added to Page 5

Future key research priorities should include, for example, stratified randomised trials and responder analyses by baseline deficiency.

  1. Clarify leucine threshold effects in older adults and note that leucine must usually be combined with other essential amino acids.

Change: Added to Page 15

Due to anabolic resistance in older adults, approximately twice as much dietary leucine than young adults is required to achieve similar increases in MPS [140-142]. Also, other AAs usually need to be in combination with leucine to support effective MPS[143]. Females may have a lower response than males to leucine stimulation of MPS and therefore may require more leucine.  Based on fat-free mass (FFM), females may have an 11% higher leucine requirement [144].

Safety

  1. Good section; including a succinct table of frequent interactions/contraindications and a list of who should seek medical advice would be useful.

Change: new table added on Page 28

Table 2. Frequent interactions and contraindications of some nutritional supplements involved in muscle health/ homeostasis.

Nutritional Supplement

Medication

Interaction/contraindication

Magnesium

Bisphosphonates

Magnesium can decrease the absorption of medications used to treat osteoporosis. Separating consumption between oral bisphosphonates and magnesium-rich supplements, or medications by at least 2 hours is recommended [294].

Antibiotics

Magnesium can form insoluble complexes with tetracycline antibiotics, such as doxycycline as well as quinolone antibiotics, such as ciprofloxacin. These antibiotics should be taken at least 2 hours before, or 4–6 hours after, a magnesium-containing supplement [295].

Vitamin D

Orlistat

Vitamin D absorption from food and supplements can be reduced by the weight-loss drug orlistat (Xenical and alli) and a reduced-fat diet, resulting in lower 25(OH)D levels [296].

Steroids

Corticosteroid medications, such as prednisone, prescribed to reduce inflammation can reduce calcium absorption and impair vitamin D metabolism [297]

Potassium

ACE inhibitors & angiotensin receptor blockers

Treatments for hypertension and type 2 diabetes, including Angiotensin Converting Enzyme (ACE) inhibitors and angiotensin receptor blockers (ARBs) reduce urinary potassium excretion which can lead to hyperkalaemia [298, 299]

Carnitine

Pivalate-conjugated antibiotics

Carnitine interacts with pivalate-conjugated antibiotics, such as pivampicillin, that are used to prevent urinary tract infections [300]. Chronic administration of these antibiotics can lead to carnitine depletion. However, although tissue carnitine levels in people who take these antibiotics may become low enough to limit fatty acid oxidation, no cases of illness due to carnitine deficiency in this population have been described [301]

Omega-3

Warfarin (Coumadin) and similar anticoagulants

Omega-3 has antiplatelet effects at high doses, and might prolong clotting times, when it is taken with warfarin (Jalili et al., 2007) However, the risk of clinically significant bleeding is not impacted or produced by omega-3s [302, 303]

  1. Add a short note on third-party testing and variability of labels to address concerns about product quality.

Changes Pages 26-27

There is a paucity of systematic studies of adverse effects [287]. Their regulation differs from prescription and over the counter products. Dietary supplements can be brought to the market without the support of clinical trials [287]. There is no requirement for demonstrating the efficacy for any health condition, however manufacturers are not allowed to claim that the supplement can be used for treating or preventing any particular disease [287]

Manufacturers and suppliers are responsible for composition, purity and strength of their products and being able to demonstrate safety. The label claims must be truthful and not misleading and contain details of the recommended dose and all ingredients including fillers, binders, and flavourings. Unfortunately, quality control and adequate labelling are sometimes lacking among the numerous products available globally[288, 289]. Tests performed by independent third-party organisations are currently needed to mitigate concerns regarding product quality and verify the safety of nutritional supplements [290, 291].

Third-party testing programs may assist in regulating the purity and safety of supplements, which entails independent third-party organisations performing specific tests verifying the safety of nutritional supplements [291]. Examples of third-party testing organizations include the NSF Certified for Sport (National Sanitation Foundation International, Ann Arbor. MI, USA) and Informed Sport (LCG, Teddington, UK), which test every batch of products before market release and attests products do not contain substances banned by sports organizations, supplement contents match printed labels, and unsafe and doping-related contaminants are not detectable [292].

Conclusion

The conclusion is well, cautiously so. Add one last sentence that supplements must complement rather than replace exercise and a high-quality diet, and that testing for deficiencies should be done before supplementation.

Change Page 32

If taking a supplement is considered potentially beneficial, to complement rather than replace exercise and a high-quality diet, then administration should ideally be under medical supervision. Where possible, this should follow a body composition and nutritional deficiency assessment and be tailored to the individual’s current lifestyle, health issues and medication.

Overall the paper needs:

(i) clearer reporting of the literature search and study selection

(ii) explicit appraisal of evidence quality (e.g., a strength-of-evidence table or GRADE-style summary)

(iii) greater consistency in terminology and scope

(iv) tighter alignment between the article type (narrative review) and statements in the back matter (which currently say “scoping review”).

The declarations are appropriately transparent but, given multiple industry affiliations, the funder’s role needs to be clarified beyond “arm’s length.”

Reviewer 3 Report

Comments and Suggestions for Authors

General comments
The aims of the presented study was twofold, to: (1) examine the pathophysiology of muscle function through a 64 nutritional lens, (2) to look at the potential benefits and harms of some com-65 monly proposed dietary supplements in non-athlete adults. Generally, the paper is well-written, easy to follow and adds merit to the scientific area of interests. However, I suggest some major improvements. Most of them are methodological flaws. All should be addressed before considering this work to be published.

Title
The title is clear, precisely inform about the meritum and the way of analysis (narrative review).

Abstract
Abstract is well-prepared

Introduction 
Aim of the work
he aim of the review is poorly justified; it is unclear why this particular review is needed.I have two main suggestons:
1) the novelty of the paper is not well defined — it is not clear what gap in the existing knowledge this article intends to fill,
2) the rationale should be stronger and more multifaceted, clearly explaining both the scientific and practical significance of the review.

Methodology
The scope and selection of the literature are not sufficiently broad. The authors should include more interregional studies and increase the proportion of works published within the last 3–5 years.
Furthermore, the authors did not emphasize different perspectives and controversies enough, and they failed to clearly highlight the most important aspects from their own point of view.\
While the literature appears to be selected purposefully rather than randomly — which is a notable strength — the manuscript lacks a section that briefly characterizes the general relationships between dietary behaviors and physical activity, to then smoothly transition to the discussion of dietary behaviors related to supplementation, physical activity, and their impact on muscle quality
I recommend adding in Section 3. Background a new subsection titled:
3.7 Relationship between Physical Activity and Dietary Behavior Patterns with the Supplementation Aspect
This subsection could include the following references:
1. Domaradzki, J. Congruence between Physical Activity Patterns and Dietary Patterns Inferred from Analysis of Sex Differences in Lifestyle Behaviors of Late Adolescents from Poland: Cophylogenetic Approach. Nutrients 2023, 15, 608. https://doi.org/10.3390/nu15030608
2. Pellegrini, M.; Ponzo, V.; Rosato, R.; Scumaci, E.; Goitre, I.; Benso, A.; Belcastro, S.; Crespi, C.; De Michieli, F.; Ghigo, E.; et al. Changes in Weight and Nutritional Habits in Adults with Obesity during the “Lockdown” Period Caused by the COVID-19 Virus Emergency. Nutrients 2020, 12, 2016. https://doi.org/10.3390/nu12072016
3. Alasqah, I.; Mahmud, I.; East, L.; Usher, K. Patterns of Physical Activity and Dietary Habits among Adolescents in Saudi Arabia: A Systematic Review. Int. J. Health Sci. (Qassim) 2021, 15(2), 39–48. PMID: 33708043; PMCID: PMC7934132.
Lombardo, M.; Feraco, A.; Camajani, E.; Gorini, S.; Strollo, R.; Armani, A.; 4. Padua, E.; Caprio, M. Effects of Different Nutritional Patterns and Physical Activity on Body Composition: A Gender and Age Group Comparative Study. Foods 2024, 13, 529. https://doi.org/10.3390/foods13040529

The structure and organization of the manuscript are appropriate and logically coherent. The content follows a clear thematic progression, with each section having a distinct purpose that guides the reader through the argumentation in a consistent and comprehensible way. The text maintains continuity without unnecessary repetitions or digressions, which contributes to its readability and clarity.

However, despite the very thorough analysis of dietary supplements, the manuscript lacks an equally detailed examination of physical activity. This is particularly important, as the authors included in the title the phrase "Dietary Supplements Support Muscle Function and Physical Activity." There is no sufficiently comprehensive discussion of studies addressing the relationship between supplementation and physical activity. Moreover, it remains unclear whether supplementation supports physical activity, or whether physical activity itself creates the demand for supplementation. These relationships are bidirectional, and this aspect should be clearly described and discussed in the manuscript.

Discussion
Discussion is well-prepared, however should be supplemented with new parts added to main text (PA and Dietary behawior Patterns)

Conclusions
IIn contrast to the discussion section, the conclusions are relatively weak. They are overly long and largely repeat content already presented earlier in the manuscript. The conclusions should instead summarize the key findings in a concise and balanced manner, emphasizing the main syntheses reached by the authors without being overly general or superficial.

General comments:
The style and language quality of the manuscript are very good. The text is written in a clear, coherent, and precise manner, appropriate for a scientific audience. The terminology is used consistently and correctly throughout the paper, and the authors successfully avoid unnecessary jargon or unsupported statements. Moreover, the referencing and formatting appear to be in full compliance with the journal’s guidelines, which enhances the overall professionalism and readability of the manuscript.

The overall impression of the manuscript is positive. The topic is relevant and valuable for the readership of the journal, both from a theoretical and practical perspective. However, certain sections require clarification and further development to strengthen the scientific contribution and coherence of the paper. After addressing the highlighted issues and refining the indicated aspects, I would be able to confidently recommend the manuscript for acceptance after major substantive and editorial revisions.

Author Response

Manuscript ID: nutrients-3948215

Type of manuscript: Narrative review

Title: Can dietary supplements support muscle function and physical activity?

A narrative review

Authors: Louise Brough *, Gail A Rees *, Lylah Drummond-Clarke, Jennifer E. McCallum, Elisabeth Taylor, Oleksii Kozhevnikov, Steven Walker

Response to Reviewer 3

(previously Reviewer 2)

We are grateful to the reviewer for the positive comments and helpful suggestions. These have served to make a better paper.

This is intended to be an unbiased scientific article. Numerous dietary supplements are discussed in a neutral manner. In controversial areas where evidence and opinions may differ, we have selected key references in conjunction with the lead author experts to hopefully provide balance. While we did not formally evaluate the quality of each paper in this huge work, most would probably be considered weak.

General comments

The aims of the presented study was twofold, to: (1) examine the pathophysiology of muscle function through a 64 nutritional lens, (2) to look at the potential benefits and harms of some com-65 monly proposed dietary supplements in non-athlete adults. Generally, the paper is well-written, easy to follow and adds merit to the scientific area of interests. However, I suggest some major improvements. Most of them are methodological flaws. All should be addressed before considering this work to be published.

Title
The title is clear, precisely inform about the meritum and the way of analysis (narrative review).

Abstract
Abstract is well-prepared

Introduction 
Aim of the work
he aim of the review is poorly justified; it is unclear why this particular review is needed.I have two main suggestons:

1) the novelty of the paper is not well defined — it is not clear what gap in the existing knowledge this article intends to fill,
2) the rationale should be stronger and more multifaceted, clearly explaining both the scientific and practical significance of the review.

Addition to Page 3:

Questions remain regarding the general population who are largely sedentary or only occasionally involved in sport. These groups are underrepresented in current nutritional research. Rather than supplementing to maximise athletic performance, what evidence exists of their efficacy in supporting muscle function and recovery in subgroups such as peri- and menopausal women, the ill and elderly? Since these groups experience the greatest muscle loss and functional decline, this evaluation is of scientific importance. Practically, this review hopes to inform clinicians, policymakers and consumers to make informed, evidence-based choices in an expanding and largely unregulated supplement market.

Clearly many individuals consider dietary supplementation to be of value: sales of single and combination products continue to rise The value of the global dietary supplements market was estimated to be worth nearly USD 152 billion in 2021[4]. It is expected to expand at an annual rate of 8.9% until 2030, notably in North America [5].

Methodology
The scope and selection of the literature are not sufficiently broad. The authors should include more interregional studies and increase the proportion of works published within the last 3–5 years.

Response/change

While the MS has been under review, we have added >25 new references that we believe has broadened the scope of the literature.

Furthermore, the authors did not emphasize different perspectives and controversies enough, and they failed to clearly highlight the most important aspects from their own point of view.\

Our aim was to provide a balanced perspective throughout this review, where the data has been weak, limited, or equivocal, we have communicated this while being careful not to perpetuate controversial perspectives.

While the literature appears to be selected purposefully rather than randomly — which is a notable strength — the manuscript lacks a section that briefly characterizes the general relationships between dietary behaviors and physical activity, to then smoothly transition to the discussion of dietary behaviors related to supplementation, physical activity, and their impact on muscle quality

I recommend adding in Section 3. Background a new subsection titled:
3.7 Relationship between Physical Activity and Dietary Behavior Patterns with the Supplementation Aspect

Change:

Good advice, See new section below

This subsection could include the following references:

  1. Domaradzki, J. Congruence between Physical Activity Patterns and Dietary Patterns Inferred from Analysis of Sex Differences in Lifestyle Behaviors of Late Adolescents from Poland: Cophylogenetic Approach. Nutrients 2023, 15, 608. https://doi.org/10.3390/nu15030608
    2. Pellegrini, M.; Ponzo, V.; Rosato, R.; Scumaci, E.; Goitre, I.; Benso, A.; Belcastro, S.; Crespi, C.; De Michieli, F.; Ghigo, E.; et al. Changes in Weight and Nutritional Habits in Adults with Obesity during the “Lockdown” Period Caused by the COVID-19 Virus Emergency. Nutrients 2020, 12, 2016. https://doi.org/10.3390/nu12072016
    3. Alasqah, I.; Mahmud, I.; East, L.; Usher, K. Patterns of Physical Activity and Dietary Habits among Adolescents in Saudi Arabia: A Systematic Review. Int. J. Health Sci. (Qassim) 2021, 15(2), 39–48. PMID: 33708043; PMCID: PMC7934132.
    Lombardo, M.; Feraco, A.; Camajani, E.; Gorini, S.; Strollo, R.; Armani, A.; 4. Padua, E.; Caprio, M. Effects of Different Nutritional Patterns and Physical Activity on Body Composition: A Gender and Age Group Comparative Study. Foods 2024, 13, 529. https://doi.org/10.3390/foods13040529

3.7 The relationship between physical activity, dietary behaviour and supplementation

Growing evidence demonstrates a robust relationship between physical activity and dietary behaviour that is bidirectional, underscoring the importance of both for optimal health. Principal Component Analysis revealed an alignment of healthy dietary habits with increased physical activity that is mutually reinforcing, suggesting that integrated lifestyle interventions can enhance population health and quality of life [66]. Patterns of supplement use closely mirror regular engagement in physical activity, with individuals maintaining consistent activity levels over time more likely to use dietary supplements. Moreover, those motivated by athletic performance or competition are significantly more inclined toward supplementation [67].

Dietary choices and physical activity not only affect disease risk, but also body composition. Evidence from comparative studies reveal that active individuals tend to select diets higher in fibre and lower in fat, with vegetarians typically demonstrating lower body mass’ and visceral adiposity than omnivores, even without the inclusion of sport. Flexitarian dietary patterns appear particularly advantageous for weight management among women, reinforcing the case for tailored nutritional recommendations [68]. Furthermore, adverse lifestyle conditions such as reduced activity, increased consumption of unhealthy foods, and psychological stressors were found to contribute significantly to weight gain during periods of environmental stress, as observed in the COVID-19 lockdown [69]. Collectively, these findings suggest that synergistic modifications in both diet and physical activity are pivotal for achieving favourable body composition and improved health, while supplementation practices are shaped by physical activity, consistency, and individual motivation.

The structure and organization of the manuscript are appropriate and logically coherent. The content follows a clear thematic progression, with each section having a distinct purpose that guides the reader through the argumentation in a consistent and comprehensible way. The text maintains continuity without unnecessary repetitions or digressions, which contributes to its readability and clarity.

However, despite the very thorough analysis of dietary supplements, the manuscript lacks an equally detailed examination of physical activity. This is particularly important, as the authors included in the title the phrase "Dietary Supplements Support Muscle Function and Physical Activity." There is no sufficiently comprehensive discussion of studies addressing the relationship between supplementation and physical activity. Moreover, it remains unclear whether supplementation supports physical activity, or whether physical activity itself creates the demand for supplementation. These relationships are bidirectional, and this aspect should be clearly described and discussed in the manuscript.

Addition to new Section 3.7

Similarly, dietary supplementation and physical activity synergistically enhance the effectiveness of the other to support muscle function and overall physiological health. Supplements such as calcium, vitamin D, omega-3 fatty acids, and whey protein have demonstrated greater efficacy when combined with consistent exercise, contributing to improvements in muscle mass, bone density, and reduced risk of sarcopenia in middle-aged and older women [70]. Structured training regimens are necessary to maximise the anabolic potential of specific supplements, particularly proteins and select botanicals [70,71]. Importantly, regular physical activity not only supports supplement efficacy, but also fosters a physiological demand for additional nutritional support, highlighting the need for individualised assessment and targeted integration of supplements within the context of an active lifestyle.

Discussion
Discussion is well-prepared, however should be supplemented with new parts added to main text (PA and Dietary behawior Patterns)

Change:

We have added new text to Page 31

We have demonstrated that physical activity and dietary habits, as well as physical activity and supplement use, share a bidirectional and mutually reinforcing relationship, with evidence showing that each behaviour can enhance the effects and uptake of the other [70, 71].

Conclusions
IIn contrast to the discussion section, the conclusions are relatively weak. They are overly long and largely repeat content already presented earlier in the manuscript. The conclusions should instead summarize the key findings in a concise and balanced manner, emphasizing the main syntheses reached by the authors without being overly general or superficial.

Response

This whole section has been revised shortened

Dietary supplementation can be of benefit to optimise muscle function, performance, mass/strength and ADL, reduce loss and aid recovery in non-athletes. Only limited subgroups have been studied, notably athletic population. Unfortunately, studies are lacking where they are needed: in peri- and menopausal women, the immobile, and the elderly where sarcopenia is an increasing issue. Combining multi-nutrient supplementation with exercise to maximise the anabolic potential is considered important. However, both detecting a nutritional deficit at cellular level and demonstrating a positive clinical effect of supplementation is difficult outside of a laboratory setting. For some there may be a psychological benefit from dietary supplementation.

While favourable data (and any evidence of harm) from taking supplements are inconsistent and of variable quality, these should not be dismissed as our knowledge of inflammation modification, the microbiome and gut muscle axis continues to evolve. The benefits of additional protein, vitamin D and the electrolytes potassium and magnesium seem widely accepted. Other supplements highlighted in this article all have their supporters: creatine, leucine, HMB, L-carnitine, turmeric, n-3 PUFAs and pre and probiotics. Most are safe in recommended doses, though interactions can occur. For some, co-ingestion and timing may be important. As so often in complex interrelated systems, more research is needed across the whole field of dietary supplementation.

If taking a supplement is considered potentially beneficial, to complement rather than replace exercise and a high-quality diet, then administration should ideally be under medical supervision. Where possible, this should follow a body composition and nutritional deficiency assessment and be tailored to the individual’s current lifestyle, health issues and medication. Unfortunately, most individuals will continue to consume dietary supplements without any involvement from a HCP. Because maintenance of skeletal muscle depends on a range of anabolic and catabolic factors, they should be encouraged to ingest a multicomponent product from an established manufacturer with adherence to dosing recommendations.

General comments:
The style and language quality of the manuscript are very good. The text is written in a clear, coherent, and precise manner, appropriate for a scientific audience. The terminology is used consistently and correctly throughout the paper, and the authors successfully avoid unnecessary jargon or unsupported statements. Moreover, the referencing and formatting appear to be in full compliance with the journal’s guidelines, which enhances the overall professionalism and readability of the manuscript.

The overall impression of the manuscript is positive. The topic is relevant and valuable for the readership of the journal, both from a theoretical and practical perspective. However, certain sections require clarification and further development to strengthen the scientific contribution and coherence of the paper. After addressing the highlighted issues and refining the indicated aspects, I would be able to confidently recommend the manuscript for acceptance after major substantive and editorial revisions.

Round 2

Reviewer 2 Report

Comments and Suggestions for Authors

I sincerely thank the authors for their thorough and thoughtful revisions. The revised manuscript demonstrates substantial improvement, with clear responses to previous comments and enhanced scientific clarity. The work now meets the standards of a scholarly publication and provides meaningful contributions to the field. I recommend the manuscript for acceptance, pending the final decision of the Academic Editor.

Reviewer 3 Report

Comments and Suggestions for Authors

I accept the manuscript in present form.